

# Conformal field theories
# on deformed spheres, anomalies, and supersymmetry

**Joseph A. Minahan[*], Usman Naseer[†] and Charles Thull[‡]**

Department of Physics and Astronomy, Uppsala University,
Box 516, SE-751 20 Uppsala, Sweden

[*] joseph.minahan@physics.uu.se, [†] usman.naseer@physics.uu.se, [‡] charles.thull@physics.uu.se

## Abstract

We study the free energy of four-dimensional CFTs on deformed spheres. For generic nonsupersymmetric CFTs only the coefficient of the logarithmic divergence in the free energy is physical, which is an extremum for the round sphere. We then specialize to $\mathcal{N} = 2$ SCFTs where one can preserve some supersymmetry on a compact manifold by turning on appropriate background fields. For deformations of the round sphere the $c$ anomaly receives corrections proportional to the supersymmetric completion of the $(\text{Weyl})^2$ term, which we determine up to one constant by analyzing the scale dependence of various correlators in the stress-tensor multiplet. We further show that the double derivative of the free energy with respect to the marginal couplings is proportional to the two-point function of the bottom components of the marginal chiral multiplet placed at the two poles of the deformed sphere. We then use anomaly considerations and counter-terms to parametrize the finite part of the free energy which makes manifest its dependence on the Kähler potential. We demonstrate these results for a theory with a vector multiplet and a massless adjoint hypermultiplet using results from localization. Finally, by choosing a special value of the hypermultiplet mass where the free energy is independent of the deformation, we derive an infinite number of constraints between various integrated correlators in $\mathcal{N} = 4$ super Yang-Mills with any gauge group and at all values of the coupling, extending previous results.



## 1. Introduction and summary

The free energy of a conformal field theory on a compact four-manifold $\mathcal{M}$ is ambiguous due to ultraviolet divergences. These are classified by diffeomorphism invariant local counter-terms of dimension four or less. The general answer for the free energy on $\mathcal{M}$ is

$$\log Z_{\mathcal{M}} = A_1 \left(\text{vol}_{\mathcal{M}} \Lambda_{\text{UV}}^4\right) + A_2 \left(\text{vol}_{\mathcal{M}} \Lambda_{\text{UV}}^4\right)^{\frac{1}{2}} + A_0 \log \left(\text{vol}_{\mathcal{M}} \Lambda_{\text{UV}}^4\right) + \text{finite}. \tag{1.1}$$

The coefficients of divergent terms as well as the finite term may depend on various parameters in the theory such as marginal couplings and the number of degrees of freedom. The quartic and quadratic divergences correspond to cosmological constant and Einstein-Hilbert counter-terms respectively. Due to the logarithmic divergence, the finite part of the free energy is scheme-dependent.

The coefficient of the logarithmic divergence is meaningful and is related to the conformal anomaly [1–4]. In four dimensions the conformal anomaly is comprised of two terms, the $a$ anomaly coming from the integrated Euler density, and the $c$ anomaly from an integrated (Weyl)$^2$ term. Their contribution to the action is

$$A_0 = \frac{1}{64\pi^2} \int d^4 x \sqrt{g} \left(-a E_4 + c\, C_{\mu\nu\rho\sigma} C^{\mu\nu\rho\sigma}\right). \tag{1.2}$$

The $a$ and $c$ anomalies belong to different classes, "type-A" and "type-B" anomalies [5]. The type-A anomalies can be expressed in terms of topological invariants and do not change under small deformations of the metric and other background fields. Anomalies in this class are monotonic under RG flows to the IR [6]. On the other hand, the type-B anomalies are not topological invariants and can be related to correlators of local operators. In particular, $c$ is related to the normalization of the stress-tensor two-point function, $C_T$, by $c = \frac{C_T}{160}$ [7,8].

If $\mathcal{M}$ is the round-sphere then the (Weyl)$^2$ term is zero and the $c$ anomaly does not contribute. Hence, one finds for $A_0$

$$A_0 = -\frac{a}{64\pi^2} \int d^4 x \sqrt{g} E_4 = -a. \tag{1.3}$$

If we deform away from the round sphere the $a$ anomaly does not change, but the (Weyl)$^2$ term is no longer zero and contributes to the free energy and $A_0$. Since the stress-tensor couples to the deformations of the metric, the change in $A_0$ is computable from the integrated correlation

functions of the stress-tensor. To leading order the change comes from the integrated stress-tensor two-point function.

One can also study a conformal field theory on $\mathcal{M}$ in the presence of other background fields for various conserved currents. In particular, if we have an $\mathcal{N} = 2$ superconformal field theory (SCFT), then in order to preserve supersymmetry we must turn on other background fields in the supergravity multiplet when deforming the metric [9–11]. This leads to additional contributions to the conformal anomaly, which results in the supersymmetric completion of the Euler density and the (Weyl)$^2$ terms. The supersymmetric Euler density includes the second Chern class of the background gauge fields and preserves the topological nature of the $a$ anomaly [12]. By invoking Weyl invariance and $R$-symmetry invariance, we further determine the supersymmetric completion of the (Weyl)$^2$ term up to six overall coefficients. We then study the scale dependence of the two-point functions of the tensor multiplet to fix all but one coefficient. This generalizes the result in [13, 14] for the supersymmetrized Weyl anomaly by including the contribution of all background fields in the supergravity multiplet. The sixth coefficient requires knowledge of the three- and four-point functions to completely fix it and will not be considered in this paper.

Extending to an $\mathcal{N} = 2$ SCFT also restricts many of the counterterms and leads to some scheme independent finite terms. Of particular interest is the dependence of the free energy on the marginal couplings in the theory. In [15, 16] it was shown that the sphere free energy of $\mathcal{N} = 2$ SCFTs is proportional to the Kähler potential. Using localization we generalize a particular version of this result to *any* supersymmetric background. Namely, we show that for the deformed sphere

$$\partial_i \partial_{\bar{j}} \log Z_{\mathcal{M}} = (32r^2)^2 \langle A_i(N) \overline{A}_{\bar{j}}(S) \rangle, \tag{1.4}$$

where $A_i$ is the bottom component of the exactly marginal chiral multiplet. (N) and (S) denote the north and south poles on the deformed sphere, which are defined as fixed points of the Killing vector composed from a preserved supersymmetry transformation. For an arbitrary supersymmetric background, the Killing vector can have more than two fixed points and the result generalizes by including a sum over the fixed points (see eq. (3.40)).

The two-point function appearing in (1.4) is proportional to the Zamolodchikov metric due to the supersymmetry. We then combine this result with the moduli anomaly [13, 14] and an analysis of possible counterterms [17] to further constrain the form of the free energy on general manifolds. We show that up to holomorphic functions and terms local in the supergravity background fields, the free energy takes the form

$$\log Z = \frac{K(\tau_i, \overline{\tau}_i)}{12} + \frac{\alpha}{96} K(\tau_i, \overline{\tau}_i) I_{(\text{Weyl})^2} + \frac{1}{96} \beta(\tau_i, \overline{\tau}_i) I_{(\text{Weyl})^2} + \gamma(\tau_i, \overline{\tau}_i, b) + P_h(\tau_i, b) + \overline{P}_h(\overline{\tau}_{\bar{i}}, b), \tag{1.5}$$

where $\alpha$ is a constant, $\beta(\tau_i, \overline{\tau}_i)$ and $\gamma(\tau_i, \overline{\tau}_i, b)$ are modular-invariant, and $P_h$ and $\overline{P}_h$ are holomorphic and anti-holomorphic functions of the moduli. $\gamma, P_h$ and $\overline{P}_h$ are also Weyl-invariant and necessarily non-local functionals of the supergravity background fields. $b$ parameterizes the deformation away from the round sphere. We then show that the partition function of the theory with a vector multiplet and an adjoint massless hypermultiplet on a specific deformed background [18], which can be computed exactly using localization, indeed has the structure of eq. (1.5).

We finally point out that the deformation independence of the free energy of the theory with a special value of hypermultiplet mass can be used to obtain an infinite number of relations between various integrated correlators at all values of the coupling. Two of these constraints were recently obtained by studying the free energy of $\mathcal{N} = 2^*$ theory on the deformed sphere [19].

The rest of the paper is structured as follows. In section 2 we review the extraction of the Weyl anomaly from the stress-tensor two-point function. In section 3 we generalize this

to $\mathcal{N} = 2$ SCFTs and compute the supersymmetric Weyl anomaly up to one undetermined constant. We then study the dependence of $\log Z$ on the marginal couplings of the SCFT and derive the results in eqs. (1.4) and (1.5). In section 4 we continue our study of $\mathcal{N} = 2$ theories on the ellipsoid. We compute the localized partition function for a gauge theory with an adjoint hypermultiplet. We consider both the $U(1)$ case and that of $SU(N)$ at large $N$. We then discuss the ambiguities of defining the theory away from $S^4$ where the space is no longer conformally flat. Finally we derive an infinite number of constraints for integrated correlators in $\mathcal{N} = 4$ SYM on the round sphere from the partition function of the deformed sphere. In the appendix we derive Ward identities for two-point functions.

## 2. CFTs on deformed spheres and the stress-tensor two-point functions

The deformations of the free energy with respect to the background metric yield correlators involving the stress-tensor. Since the Euler invariant does not change under metric perturbations, only the (Weyl)$^2$ term contributes. If we denote the perturbation away from the round four-sphere metric by $h_{\mu\nu}$, i.e.,

$$\mathrm{d}s^2_{\text{deformed}} = \mathrm{d}s^2_{\text{round}} + h_{\mu\nu}\mathrm{d}x^\mu \mathrm{d}x^\nu = \Omega^2 \delta_{ab}\mathrm{d}x^a\mathrm{d}x^b + h_{\mu\nu}\mathrm{d}x^\mu\mathrm{d}x^\nu, \tag{2.1}$$

where $\Omega = \frac{2}{1+\frac{|x|^2}{r^2}}$ and $r$ is the radius of the round sphere, then the leading contribution of the (Weyl)$^2$ term to the anomaly $A_0$ is given by

$$\delta A_0 = \frac{c}{256\pi^2} \int \mathrm{d}^4x \sqrt{g} h^{\mu\nu} \left( \pi_{\mu\rho}\pi_{\nu\sigma} + \pi_{\mu\sigma}\pi_{\nu\rho} - \frac{2}{3}\pi_{\mu\nu}\pi_{\rho\sigma} \right) h^{\rho\sigma}, \tag{2.2}$$

where $\pi_{\mu\nu} \equiv \nabla_\mu\nabla_\nu - g_{\mu\nu}\nabla^2$. The combination in the parentheses projects to traceless, transverse, rank-two tensors. To relate the above expression to the stress-tensor two-point correlator, we use the fact that the integral is invariant under the Weyl scaling of the full metric $g_{\mu\nu} + h_{\mu\nu}$. Scaling the metric by $\Omega^{-2}$ and keeping only the leading term in the free energy we get

$$\delta A_0 = \frac{c}{256\pi^2} \int \mathrm{d}^4x \Omega^2(x) h^{\mu\nu} \left( \pi_{\mu\rho}\pi_{\nu\sigma} + \pi_{\mu\sigma}\pi_{\nu\rho} - \frac{2}{3}\pi_{\mu\nu}\pi_{\rho\sigma} \right) \Omega^2(x) h^{\rho\sigma}(x), \tag{2.3}$$

where the operators $\pi_{\mu\nu}$ are those for the flat metric. To simplify the above expression further, we introduce an integral over a $\delta$-function followed by an integration by parts to get

$$\delta A_0 = \frac{c}{256\pi^2} \int \mathrm{d}^4x \int \mathrm{d}^4y \Omega^2(x)\Omega^2(y) h^{\mu\nu}(x) h^{\rho\sigma}(y)$$
$$\times \left( \pi_{\mu\rho}\pi_{\nu\sigma} + \pi_{\mu\sigma}\pi_{\nu\rho} - \frac{2}{3}\pi_{\mu\nu}\pi_{\rho\sigma} \right) \delta^4(x-y). \tag{2.4}$$

We can further manipulate (2.4) by using the following regularization procedure to define the Dirac delta function [20–22][1],

$$\delta^4(x) = \frac{-1}{2V_{\mathbb{S}^3}}\nabla^2\frac{1}{|x|^2} = \frac{-1}{2V_{\mathbb{S}^3}}\nabla^2\delta_\sigma\frac{\log(|x|\Lambda_{\text{UV}})}{|x|^2} = \frac{1}{V_{\mathbb{S}^3}}\delta_\sigma\frac{1}{|x|^4}, \tag{2.5}$$

where the first and the second equations hold identically, while the last is true away from $|x| = 0$ and $\delta_\sigma = \frac{\mathrm{d}}{\mathrm{d}\log\Lambda_{\text{UV}}}$ captures the dependence on the scale. A short calculation then gives

$$\left( \pi_{\mu\rho}\pi_{\nu\sigma} + \pi_{\mu\sigma}\pi_{\nu\rho} - \frac{2}{3}\pi_{\mu\nu}\pi_{\rho\sigma} \right) \frac{1}{|x-y|^4} = 640\Omega^2(x)\Omega^2(y)\frac{\mathcal{I}_{\mu\nu\rho\sigma}(x,y)}{s(x,y)^8}, \tag{2.6}$$

---

[1]This amounts to regularizing the coincident limit of the two-point functions. This regularization introduces a length scale and the type-B conformal anomaly is due to the dependence of free energy on this length scale [5].

where $\mathcal{I}_{\mu\nu\rho\sigma}$ is the tensor structure appearing in the two point function

$$\langle T_{\mu\nu}(x)\,T_{\rho\sigma}(y)\rangle = \frac{C_T}{V_{\mathbb{S}^{d-1}}^2}\frac{\mathcal{I}_{\mu\nu,\rho\sigma}(x,y)}{s(x,y)^{2d}}, \tag{2.7}$$

$s(x,y)$ is the geodesic distance on the sphere

$$s(x,y) = \sqrt{\Omega(x)\Omega(y)}|x-y|, \tag{2.8}$$

and

$$\mathcal{I}_{\mu\nu,\rho\sigma}(x) = \frac{1}{2}\left(I_{\mu\sigma}(x)I_{\nu\rho}(x) + I_{\mu\rho}(x)I_{\nu\sigma}(x)\right) - \frac{1}{d}g_{\mu\nu}g_{\rho\sigma}, \tag{2.9}$$

with

$$I_{\mu\nu}(x-y) = \delta_{\mu\nu} - 2\frac{(x-y)_\mu(x-y)_\nu}{|x-y|^2}. \tag{2.10}$$

Plugging everything in, we get

$$\delta A_0 = \frac{1}{32}\delta_\sigma \int \mathrm{d}^4x\sqrt{g(x)}\int \mathrm{d}^4y\sqrt{g(y)}h^{\mu\nu}(x)h^{\rho\sigma}(y)\langle T_{\mu\nu}(x)T_{\rho\sigma}(y)\rangle. \tag{2.11}$$

Hence, the leading correction to the universal coefficient in the free energy is given by the integrated stress-tensor two-point function.

### 2.1. Examples

Let us demonstrate the above by considering generic CFTs placed on specific deformed spheres.

#### 2.1.1. $SU(2)\times U(1)$ isometry

We first consider a simple deformation which preserves an $SU(2)\times U(1)$ isometry. In projective coordinates the deformation is

$$h_{\mu\nu}\mathrm{d}x^\mu\mathrm{d}x^\nu = \varepsilon\Omega^4(x_2\mathrm{d}x_1 - x_1\mathrm{d}x_2)^2. \tag{2.12}$$

The leading contribution to the (Weyl)$^2$ part of the anomaly is then given by

$$\delta A_0 = \frac{c}{64\pi^2}\int \mathrm{d}^4x\sqrt{g}C_{\mu\nu\rho\sigma}C^{\mu\nu\rho\sigma} = \frac{3\varepsilon^2}{2240}C_T. \tag{2.13}$$

Let us now compute the logarithmic divergence in the integrated stress-tensor two-point function. Contracting the correlator with the metric deformation we have

$$\begin{aligned}
&\sqrt{g(x)}\sqrt{g(y)}h^{\mu\nu}(x)h^{\rho\sigma}(y)\langle T_{\mu\nu}(x)T_{\rho\sigma}(y)\rangle\\
&= \frac{\varepsilon^2 C_T\Omega(x)^2\Omega(y)^2}{16\pi^4|x-y|^8}\left(4(x_1y_1 + x_2y_2)^2 - \left(x_1^2 + x_2^2\right)\left(y_1^2 + y_2^2\right)\right.\\
&\left. + 16\frac{(x_2y_1 - x_1y_2)^2}{|x-y|^2}\left(\frac{(x_2y_1 - x_1y_2)^2}{|x-y|^2} - (x_1y_1 + x_2y_2)\right)\right).
\end{aligned} \tag{2.14}$$

In general, to compute the integrated correlator one can use the $SO(5)$ symmetry of the integration measure to fix the position of one of the operators at the north (or south) pole. This corresponds to a specific choice of regularization scheme which preserves the $SO(5)$ isometry of the round sphere. If we do this the above correlator vanishes identically at separated points

because the deformation is zero at the poles. To uncover the singularities in the coincident limit we use the relations

$$\delta_\sigma \frac{1}{|x|^4} = 2\pi^2 \delta^4(x), \qquad \delta_\sigma \frac{1}{|x|^{4+2m}} = \frac{2\pi^2}{4^m \Gamma(m+1)\Gamma(m+2)} \Box^m \delta^4(x-y). \tag{2.15}$$

The second equality follows from the first by interchanging the Laplacian and $\delta_\sigma$. Using these relations one finds

$$\begin{aligned}
&\frac{1}{32}\delta_\sigma \int d^4x \int d^4y \sqrt{g(x)}\sqrt{g(y)} h^{\mu\nu}(x) h^{\rho\sigma}(y) \langle T_{\mu\nu}(x) T_{\rho\sigma}(y)\rangle \\
&= -\frac{C_T \epsilon^2}{16\pi^2} \int d^4x \frac{\left(x_1^2+x_2^2\right)\left(x_1^4 + 2x_1^2\left(x_2^2+x_3^2+x_4^2-7\right)+x_2^4+2x_2^2\left(x_3^2+x_4^2-7\right)+\left(x_3^2+x_4^2+1\right)^2\right)}{\left(x_1^2+x_2^2+x_3^2+x_4^2+1\right)^8} \\
&= \frac{3 C_T \epsilon^2}{2240},
\end{aligned} \tag{2.16}$$

which matches the result in (2.13).

### 2.1.2. $U(1) \times U(1)$ isometry

Let us now consider squashing the round sphere to an ellipsoid [18]. In this case the metric is

$$ds^2 = r^2 E^a E^b \delta_{ab}, \tag{2.17}$$

where

$$E^1 = \ell \sin\rho \cos\theta \, d\phi, \qquad E^2 = \widetilde{\ell} \sin\rho \sin\theta \, d\chi, \qquad E^3 = \sin\rho f \, d\theta + h \, d\rho, \qquad E^4 = g \, d\rho. \tag{2.18}$$

The coordinates $\phi$ and $\chi$ are $2\pi$ periodic, while $\theta \in [0, \frac{\pi}{2}]$, $\rho \in [0, \pi]$, and

$$f = \sqrt{\ell^2 \sin^2\theta + \widetilde{\ell}\cos^2\theta}, \qquad g = \sqrt{r^2 \sin^2\rho + \left(\ell\widetilde{\ell}\right)^2 f^{-2}\cos^2\rho},$$

$$h = \frac{\widetilde{\ell}^2 - \ell^2}{f} \cos\rho \sin\theta \cos\theta. \tag{2.19}$$

Setting $\ell = \widetilde{\ell} = r$ corresponds to the round sphere. The overall size of the manifold is parameterized by $r^2 \ell\widetilde{\ell}$ while the squashing is parameterized by the dimensionless parameter $b = \sqrt{\frac{\ell}{\widetilde{\ell}}}$. The metric in (2.17) preserves a $U(1) \times U(1)$ isometry corresponding to the Killing vectors $\partial_\phi$ and $\partial_\chi$. For supersymmetric theories it admits a Killing spinor when certain background fields are turned on [18]. The integrated (Weyl)$^2$ term can be calculated analytically for this deformation for all $b \geq 1$ and we find

$$\begin{aligned}
&\frac{1}{16\pi^2} \int d^4x \sqrt{g} C_{\mu\nu\rho\sigma} C^{\mu\nu\rho\sigma} = \\
&\frac{-46 b^{12} + 68 b^8 - 28 b^4 + 15\sqrt{b^4-1}\, b^{10} \log\left(2b^2\left(\sqrt{b^4-1}+b^2\right)-1\right)+6}{45 b^{10}} \\
&= \mathcal{O}\left((b-1)^4\right).
\end{aligned} \tag{2.20}$$

Hence, the integrated two-point function for the stress-tensor does not have logarithmic singularities to leading order in the deformation.

We can also show the absence of a leading order singularity in the two-point function directly. We first set $\widetilde{\ell} = r$ so that the deformation is completely captured by $\ell = b^2 r$. The

deformation of the metric away from the round sphere takes the form in projective coordinates,

$$h_{\mu\nu} = 2(b-1)\left(v_\mu v_\nu + w_\mu w_\nu\right) \qquad \text{where} \qquad v_\mu dx^\mu = d\left(\Omega x^1\right) \qquad \text{and} \qquad w_\mu dx^\mu = d\left(\Omega x^2\right). \tag{2.21}$$

We now write the two-point function contracted with the deformation as

$$\sqrt{g(x)}\sqrt{g(y)}h^{\mu\nu}(x)h^{\rho\sigma}(y)\langle T_{\mu\nu}(x)T_{\rho\sigma}(y)\rangle$$
$$= \frac{C_T}{4\pi^4|x-y|^8}\Big(h^{ab}(x)h_{ab}(y) - \frac{1}{4}h^a{}_a(x)h^b{}_b(y) - 4h^{ac}(x)h_c{}^b(y)\frac{(x-y)_a(x-y)_b}{|x-y|^2}$$
$$+ 4h^{ab}(x)h^{cd}(y)\frac{(x-y)_a(x-y)_b(x-y)_c(x-y)_d}{|x-y|^4}\Big). \tag{2.22}$$

After integrating over the coordinates, the anomaly contribution of each of the four terms inside the parenthesis in (2.22) can be found by using (2.15). After a tedious calculation we find

$$\delta_\sigma \int d^4x d^4y \frac{h^{ab}(x)h_{ab}(y)}{|x-y|^8} = \int dx \frac{\pi^4\left(64x^8 + 252x^6 + 360x^4 + 202x^2 + 45\right)}{18\left(x^2+1\right)^{\frac{9}{2}}},$$

$$-\delta_\sigma \int d^4x d^4y \frac{1}{4|x-y|^8}h^a{}_a(x)h^b{}_b(y) = -\int dx \frac{\pi^4\left(12x^4 + 6x^2 - 1\right)}{24\left(x^2+1\right)^{\frac{9}{2}}},$$

$$-4\delta_\sigma \int d^4x d^4y h^{ac}(x)h_c{}^b(y)\frac{(x-y)_a(x-y)_b}{|x-y|^{10}} =$$
$$-\int dx \frac{\pi^4\left(2318x^4 + 1271x^2 + 400\left(x^2+4\right)x^6 + 278\right)}{60\left(x^2+1\right)^{\frac{9}{2}}},$$

$$4\delta_\sigma \int d^4x d^4y h^{ab}(x)h^{cd}(y)\frac{(x-y)_a(x-y)_b(x-y)_c(x-y)_d}{|x-y|^{12}}$$
$$= \int dx \frac{\pi^4\left(1120x^8 + 4560x^6 + 6888x^4 + 3676x^2 + 753\right)}{360\left(x^2+1\right)^{\frac{9}{2}}}. \tag{2.23}$$

Each of the above terms is logarithmically divergent for large $x$, but their sum vanishes. Hence, the leading logarithmic divergence for the integrated two-point function vanishes.

## 3. Free energy of $\mathcal{N}=2$ SCFTs on deformed spheres

In this section we study the partition function of $\mathcal{N}=2$ SCFTs on supersymmetric curved backgrounds. These backgrounds are obtained by coupling the stress-tensor multiplet of $\mathcal{N}=2$ theories with the gravity (Weyl) multiplet of $\mathcal{N}=2$ Poincaré (conformal) supergravity [10, 11, 23]. The supergravity background in Euclidean signature has a metric $g_{\mu\nu}$, a self-dual two-form $B_{\mu\nu}^+$, an anti-self-dual two-from $B_{\mu\nu}^-$, background vector fields $V_\mu$ and $\mathcal{V}_\mu{}^{ij}$ for $U(1)_R \times SU(2)_R$ R-symmetry and a scalar field $D(x)$. The partition function is then a non-local function of the supergravity background fields and couplings of the theory which can be computed via localization under favorable circumstances. We study both the logarithmically divergent and the finite part of the free energy. We do not need the explicit knowledge of any supersymmetric background for our analysis and our main tool is the Weyl anomaly, the moduli anomaly and a classification of local counter terms.

### 3.1. Weyl anomaly in $\mathcal{N} = 2$ SCFTs

The universal coefficient of the $\log\left(\mathrm{vol}_{\mathcal{M}}\Lambda^4_{\mathrm{UV}}\right)$ term can be determined using the Weyl anomaly which is modified to incorporate the $\mathcal{N} = 2$ supersymmetry. The appropriately supersymmetrized Weyl variation of the free energy is given by the following superspace expression [13, 14, 24].

$$\delta_{\Sigma}\log Z \supset \frac{1}{16\pi^2}\int \mathrm{d}^4x \int \mathrm{d}^4\Theta\mathcal{E}\delta\Sigma\left(a\Xi + (c-a)W^{\alpha\beta}W_{\alpha\beta}\right) + c.c. \tag{3.1}$$

Here $\delta\Sigma$ is a chiral superfield which parameterizes the super-Weyl transformations. Its lowest component is $\delta\sigma + i\,\delta\alpha$ where $\delta\sigma$ parameterizes the Weyl transformations and $\delta\alpha$ parameterizes the $U(1)_R$ transformation. $\mathcal{E}$ is the chiral density, $W_{\alpha\beta}$ is the covariantly chiral Weyl superfield and $\Xi$ is a composite scalar constructed from curvature superfields that appear in commutators of super-covariant derivatives. In component fields the anomalous variation of the free energy takes the form[2]

$$\delta_{\Sigma}\log Z \supset -2a\delta\sigma\chi(\mathcal{M}) + \delta\alpha\left[(a-c)\left(\mathcal{P}(\mathcal{M}) - n_{U(1)_R}\right) - (a - \tfrac{c}{2})n_{SU(2)_R}\right]$$
$$+ \frac{c}{16\pi^2}\delta\sigma\int \mathrm{d}^4x\sqrt{g}\left(C^{\mu\nu\rho\sigma}C_{\mu\nu\rho\sigma} + \cdots\right). \tag{3.2}$$

All terms on the first line are topological invariants, where $\chi(\mathcal{M})$ is the Euler characteristic of the compact manifold $\mathcal{M}$. The term multiplying the $U(1)_R$ transformation is written as a combination of the Pontryagin character and the second Chern class for the background gauge fields,

$$\mathcal{P}(\mathcal{M}) = \frac{1}{32\pi^2}\int \mathrm{d}^4x\epsilon^{\mu\nu\rho\sigma}R_{\mu\nu\alpha\beta}R_{\rho\sigma}{}^{\alpha\beta},$$
$$n_{U(1)_R} = \frac{1}{32\pi^2}\int \mathrm{d}^4x\epsilon^{\mu\nu\rho\sigma}F_{\mu\nu}F_{\rho\sigma}, \tag{3.3}$$
$$n_{SU(2)_R} = \frac{1}{32\pi^2}\int \mathrm{d}^4x\epsilon^{\mu\nu\rho\sigma}\mathrm{Tr}\mathcal{F}_{\mu\nu}\mathcal{F}_{\rho\sigma}.$$

For supergravity backgrounds smoothly connected to the round sphere, the topological invariants in eq. (3.3) vanish and $\chi(\mathcal{M}) = 2$. The term proportional to the central charge $c$ in the Weyl transformation is not topological and hence is non-trivial on deformed spheres. The ellipses denote the additional terms required to make the (Weyl)$^2$ term supersymmetric. Let us denote this supersymmetric completion by $\mathrm{I}_{(\mathrm{Weyl})^2}$. Then the Weyl anomaly coefficient, $A_0$, on supergravity backgrounds smoothly connected to the round sphere is given by

$$A_0 = -a + \frac{c}{64\pi^2}\mathrm{I}_{(\mathrm{Weyl})^2}. \tag{3.4}$$

Since $c$ appears in the normalization of the two-point functions for operators in the stress-tensor multiplet, this suggests that one can relate these two-point functions to the supersymmetric completion in (3.4). In the next section we follow this strategy to determine $\mathrm{I}_{(\mathrm{Weyl})^2}$.

### 3.2. Supersymmetric completion of (Weyl)$^2$ from stress-tensor correlators

In this section we determine $\mathrm{I}_{(\mathrm{Weyl})^2}$ by studying the logarithmic divergences of various two-point correlators of stress-tensor multiplet operators. We first use the Weyl-weights and

---

[2]$\delta\sigma$ and $\delta\alpha$ are independent of coordinates.

$U(1)_R$ charges of the various fields in the supergravity multiplet to write down the most general possibility for $I_{(Weyl)^2}$. We then use the precise coupling of the stress-tensor multiplet with the Weyl-multiplet to relate the logarithmic divergences of the two-point functions to various terms in $I_{(Weyl)^2}$.

The Weyl weights of the fields in the supergravity multiplet are

$$w_{g_{\mu\nu}} = -2, \qquad w_{A_\mu} = 0, \qquad w_{\mathcal{V}_\mu} = 0, \qquad w_B = -1 \qquad w_D = 2. \tag{3.5}$$

The self-dual and anti-self-dual two-forms are charged under the background $U(1)_R$ gauge field and carry opposite chiral weights. This implies that an equal number of self-dual and anti-self-dual two-forms must appear in an allowed term. Using these considerations one can list the possible local functions of background fields which can appear in $I_{(Weyl)^2}$. For example, possible terms involving the scalar field $D(x)$ are

$$g^{\mu\nu}\nabla_\mu\nabla_\nu D, \qquad D\left(B_{\mu\nu}B^{\mu\nu}\right), \qquad D^2. \tag{3.6}$$

The first term is omitted because it is a total derivative. The second term is ruled out because its non-trivial parts must involve different numbers of self-dual and anti self-dual two-forms. Similarly, after accounting for the possible terms involving other background fields one can write down the most general form for $I_{(Weyl)^2}$ consistent with the invariance under $U(1)_R$ and constant Weyl transformations:

$$I_{(Weyl)^2} = \int d^4x \sqrt{g}\Big(C_{\mu\nu\rho\sigma}C^{\mu\nu\rho\sigma} + c_1 D^2 + c_2 F_{\mu\nu}F^{\mu\nu} + c_3 \text{Tr}\mathcal{F}_{\mu\nu}\mathcal{F}^{\mu\nu} + c_4 \nabla_\mu B^{+\mu\nu}\nabla^\sigma B^-_{\sigma\nu}$$
$$+ \tilde{c}_4 R_{\mu\nu}B^{+\mu\rho}B^{-\nu}{}_\rho + c_5 B^+_{\mu\nu}B^{+\mu\nu}B^-_{\mu\nu}B^{-\mu\nu}\Big). \tag{3.7}$$

Under non-constant Weyl transformations all terms are invariant except the ones that appear with coefficients $c_4$ and $\tilde{c}_4$. The coefficient $\tilde{c}_4$ can then be fixed in terms of $c_4$ by requiring their combined Weyl variation to cancel. We rewrite these in terms of the-two form $B_{\mu\nu} = B^+_{\mu\nu} + B^-_{\mu\nu}$, such that

$$\nabla_\mu B^{+\mu\nu}\nabla^\sigma B^-_{\sigma\nu} = -\frac{1}{8}\left(\nabla_\mu B_{\nu\rho}\nabla^\mu B^{\nu\rho} - 2\nabla_\mu B^{\mu\nu}\nabla_\rho B^\rho{}_\nu - 2\nabla_\rho B^{\mu\nu}\nabla_\mu B^\rho{}_\nu\right),$$
$$R_{\mu\nu}B^{+\mu\rho}B^{-\nu}{}_\rho = \frac{1}{2}R_{\mu\nu}B^{\mu\rho}B^\nu{}_\rho - \frac{1}{8}RB^{\mu\nu}B_{\mu\nu}. \tag{3.8}$$

Up to a total derivative, the Weyl variation is then given by

$$\delta_\sigma\sqrt{g}\Big(c_4\nabla_\mu B^{+\mu\nu}\nabla^\sigma B^-_{\sigma\nu} + \tilde{c}_4 R_{\mu\nu}B^{+\mu\rho}B^{-\nu}{}_\rho\Big) = \frac{c_4 - 2\tilde{c}_4}{8}\nabla^2\sigma B^{\mu\nu}B_{\mu\nu} - \frac{c_4 - 2\tilde{c}_4}{2}\nabla_\mu\nabla_\nu\sigma B^{\mu\rho}B^\nu{}_\rho. \tag{3.9}$$

Hence, the $I_{(Weyl)^2}$ given in eq. (3.7) is invariant under local Weyl transformations if $\tilde{c}_4 = \frac{1}{2}c_4$.

The rest of the coefficients in $I_{(Weyl)^2}$, except $c_5$, can be determined by relating the Weyl anomaly to logarithmic divergences in the two-point functions. These two-point functions can be computed by taking functional derivatives of the free energy with respect to the background fields to which the operators couple. To this end we compute the scale dependence of the two-

point functions using $\delta_\sigma \log Z = 4A_0 = -4a + \frac{c}{16\pi^2} I_{(\text{Weyl})^2}$ and eq. (3.7). This gives

$$
\begin{aligned}
\delta_\sigma \frac{\delta^2 \log Z}{\delta D(x)\delta D(y)}\bigg|_{\mathbb{R}^4} &= \frac{c_1 C_T}{1280\pi^2}\delta^4(x-y), \\
\delta_\sigma \frac{\delta^2 \log Z}{\delta V_\mu(x)\delta V_\nu(y)}\bigg|_{\mathbb{R}^4} &= -\frac{c_2 C_T}{640\pi^2}\left(\delta^{\mu\nu}\partial^2 - \partial^\mu\partial^\nu\right)\delta^4(x-y), \\
\delta_\sigma \frac{\delta^2 \log Z}{\delta \mathcal{V}_\mu^{ij}(x)\delta \mathcal{V}_\nu^{kl}(y)}\bigg|_{\mathbb{R}^4} &= -\frac{c_3 C_T}{640\pi^2}\epsilon_{(ij}\epsilon_{k)l}\left(\delta^{\mu\nu}\partial^2 - \partial^\mu\partial^\nu\right)\delta^4(x-y), \\
\delta_\sigma \frac{\delta^2}{\delta B_{\mu\nu}(x)\delta B_{\rho\sigma}(y)}\log Z\bigg|_{\mathbb{R}^4} &= \frac{c_4 C_T}{10240\pi^2}\mathcal{B}_{[\mu\nu][\rho\sigma]}\delta^4(x-y),
\end{aligned}
\tag{3.10}
$$

where $\mathcal{B}_{\mu\nu\rho\sigma}$ is the differential operator

$$
\mathcal{B}_{\mu\nu\rho\sigma} = \left(\delta^{\mu\rho}\delta^{\sigma\nu}\partial^2 - 4\delta^{\mu\rho}\partial^\sigma\partial^\nu\right).
\tag{3.11}
$$

We now use the linearized coupling of the Weyl-multiplet with the stress-tensor multiplet operators [25] to relate the terms computed in (3.10) to the two-point functions, where we find

$$
\delta\mathcal{L} = \left[\frac{1}{2}h^{\mu\nu}T_{\mu\nu} - \frac{i}{2}V_\mu j^\mu - \frac{i}{2}\left(t_\mu\right)^{ij}\left(\mathcal{V}_\mu\right)_{ij} - 16\left(H_{\mu\nu}B^{+\mu\nu} + \overline{H}_{\mu\nu}B^{-\mu\nu}\right) - O_2 D\right].
\tag{3.12}
$$

The stress-tensor multiplet two-point functions are completely determined by using Ward identities in terms of the central charge $C_T$ and are given by

$$
\begin{aligned}
\langle O_2(x)O_2(y)\rangle_{\mathbb{R}^4} &= \frac{3C_T}{5120\pi^4}\frac{1}{|x-y|^4}, \\
\langle j_\mu(x)j_\nu(y)\rangle_{\mathbb{R}^4} &= -\frac{3C_T}{160\pi^4}\frac{1}{|x-y|^6}I_{\mu\nu}(x-y), \\
\langle \left(t_\mu\right)^{ij}(x)\left(t_\nu\right)^{kl}(y)\rangle_{\mathbb{R}^4} &= -\frac{3C_T}{160\pi^4}\frac{1}{|x-y|^6}I_{\mu\nu}(x-y), \\
\langle H_{\mu\nu}(x)\overline{H}_{\rho\sigma}(y)\rangle_{\mathbb{R}^4} &= \frac{3C_T}{1280\pi^4}\frac{(x-y)^\gamma(x-y)^\iota}{|x-y|^8} \\
&\quad \left(4\epsilon_{\mu\nu\gamma[\sigma}\delta_{\rho]\iota} + 4\epsilon_{\rho\sigma\iota[\nu}\delta_{\mu]\gamma} + 12\delta^{[\mu}{}_\iota\delta^\nu{}_\sigma\delta^{\gamma]}{}_\rho + 8\delta^{[\mu}{}_{[\rho}\delta_{\sigma]\iota}\delta^{\nu]\gamma}\right).
\end{aligned}
\tag{3.13}
$$

These two-point functions are derived in Appendix A.

From the linearized coupling of the background scalar we find that

$$
\delta_\sigma \frac{\delta^2 \log Z}{\delta D(x)\delta D(y)}\bigg|_{\mathbb{R}^4} = \delta_\sigma\langle O_2(x)O_2(y)\rangle = \frac{3C_T}{2560\pi^2}\delta^4(x-y).
\tag{3.14}
$$

Comparing this with (3.10) we determine that $c_1 = \frac{3}{2}$.

From the linearized coupling of the background field to the $SU(2)_R$ current we compute

$$
\delta_\sigma \frac{\delta^2 \log Z}{\delta \mathcal{V}_\mu^{ij}(x)\delta \mathcal{V}_\nu^{kl}(y)}\bigg|_{\mathbb{R}^4} = -\frac{1}{4}\delta_\sigma\langle t_{ij}^\mu(x)t_{kl}^\nu(y)\rangle.
\tag{3.15}
$$

The scale dependence of the right hand side can be computed using the two-point functions (3.13) and the identity

$$
\frac{I_{\mu\nu}}{|x|^6} = \frac{1}{12}(\delta_{\mu\nu}\partial^2 - \partial_\mu\partial_\nu)\frac{1}{|x|^4},
\tag{3.16}
$$

which gives

$$\delta_\sigma \frac{\delta^2 \log Z}{\delta \mathcal{V}_\mu^{ij}(x)\delta \mathcal{V}_\nu^{kl}(y)}\bigg|_{\mathbb{R}^4} = \frac{C_T}{1280\pi^2}\epsilon_{(ij}\epsilon_{k)l}\left(\delta^{\mu\nu}\partial^2 - \partial^\mu\partial^\nu\right)\delta^4(x-y). \tag{3.17}$$

Comparing with eq. (3.10) we get $c_3 = -\frac{1}{2}$. In a completely analogous manner one finds that $c_2 = -\frac{1}{2}$.

From the coupling of the two-form field in the Lagrangian (3.12) we find that

$$\begin{aligned}
\delta_\sigma \frac{\delta^2}{\delta B_{\mu\nu}(x)\delta B_{\rho\sigma}(y)}\log Z\bigg|_{\mathbb{R}^4} &= 256\delta_\sigma\langle H_{\mu\nu}(x)\overline{H}_{\rho\sigma}(y) + \overline{H}_{\mu\nu}(x)H_{\rho\sigma}(y)\rangle \\
&= \frac{3C_T}{5\pi^4}\delta_\sigma \frac{(x-y)^\gamma(x-y)^\iota}{|x-y|^8}\bigg(4\epsilon_{\mu\nu\gamma[\sigma}\delta_{\rho]\iota} + 4\epsilon_{\rho\sigma\iota[\nu}\delta_{\mu]\gamma} + 12\delta^{[\mu}{}_\iota\delta^\nu{}_\sigma\delta^\gamma{}_\rho \\
&\qquad + 8\delta^{[\mu}{}_{[\rho}\delta_{\sigma]\iota}\delta^{\nu]\gamma} + (\{\mu,\nu,\gamma\}\leftrightarrow\{\rho,\sigma,\iota\})\bigg).
\end{aligned} \tag{3.18}$$

Under $(\{\mu,\nu,\gamma\}\leftrightarrow\{\rho,\sigma,\iota\})$ the first two terms are antisymmetric and drop out from the two-point function. Using

$$\frac{x^\gamma x^\iota}{|x|^8} = \frac{1}{24}\left(\partial^\gamma\partial^\iota + \tfrac{1}{2}g^{\gamma\iota}\partial^2\right)\frac{1}{|x|^4}, \tag{3.19}$$

we then find for the scale dependence of the two-point function

$$\delta_\sigma \frac{\delta^2}{\delta B_{\mu\nu}(x)\delta B_{\rho\sigma}(y)}\log Z\bigg|_{\mathbb{R}^4} = \frac{C_T}{20\pi^2}\mathcal{B}_{[\mu\nu][\rho\sigma]}\delta^4(x-y). \tag{3.20}$$

Comparing with eq. (3.10) we find that $c_4 = 4096$.

The final coefficient $c_5$ is for a quartic term and hence it is necessary to compute up to four-point correlators to find it. In fact, the supersymmetric Lagrangian also contains terms coupling the bottom components of marginal chiral multiplets with $B^+_{\mu\nu}B^{+\mu\nu}$. Hence, four derivatives with respect to the background two-form field will involve a combination of two-, three- and four-point functions. In principle, all of these functions can contribute to the logarithmic divergence in the free energy.

The scale dependence of the two-point function can be ascertained as before. Scale dependence of three- and four-point functions is more non-trivial to obtain. Higher-point functions contain two types of divergences: (i) when only a subset of the operators collide, (ii) when all the operators collide. The divergences of the first kind are the so-called semi-local divergences [26] and these can be regularized by counterterms which involve coupling of the background fields for the colliding operators with the remaining operator. Since such couplings are already present in the supersymmetric Lagrangian and are completely determined by supersymmetry, the effect of such counterterms is to only renormalize the operators appearing in the Lagrangian.

The divergences of the second kind, the so-called ultra-local divergences [26] are regularized by adding counter-terms local in the background fields and these are the divergences that we are interested in. The ultra-local divergence of the three-point function can be determined with a bit of effort because the three-point functions are protected. The task becomes much more difficult for the four-point function. Since the only theory-dependent content of the Weyl anomaly is the coefficient $C_T$, we can use the free theory results to determine the ultra-local divergence in the four-point function and fix $c_5$. This would be interesting to compute but we will not do it here.

### 3.3. Finite part of the free energy and the Kähler potential

In this section we study the free energy of $\mathcal{N} = 2$ SCFTs on a supersymmetric curved background as a function of the marginal couplings. On $\mathbb{R}^4$ an $\mathcal{N} = 2$ SCFT can be deformed while preserving superconformal invariance by the term

$$\frac{1}{\pi^2} \int \mathrm{d}^4 x \sum_{i=1}^{\dim_{\mathcal{M}_C}} \left( \tau_i \mathcal{C}_i + \overline{\tau}_{\bar{i}} \overline{\mathcal{C}}_{\bar{i}} \right), \tag{3.21}$$

where $\mathcal{C}_i$ is a marginal operator in the SCFT and $\dim_{\mathcal{M}_C}$ is the dimension of the conformal manifold $\mathcal{M}_C$ of the SCFT. On a curved background the above deformation is not generally superconformal invariant. It can, however, be made superconformal invariant by adding non-minimal couplings with background fields of the supergravity Weyl-multiplet [15, 27]. This leads to a term having the form

$$\frac{1}{\pi^2} \int \mathrm{d}^4 x \sqrt{g} \sum_i \tau_i \left( \mathcal{C}_i - \frac{1}{4} \mathcal{A}_i B^+_{\mu\nu} B^{+\mu\nu} \right) + \text{h.c}, \tag{3.22}$$

where $\mathcal{A}_i$ is the bottom component of the chiral multiplet whose top component is the marginal operator $\mathcal{C}_i$. For Lagrangian SCFTs based on a gauge group $G = \prod_i G_i$, the above deformation is proportional to the action for an $\mathcal{N} = 2$ vector multiplet [28] with complexified gauge coupling $\tau_i$. For our purposes, it now suffices to focus on a single marginal deformation which has a Lagrangian description. Our results hold for abstract marginal deformations, irrespective of their microscopic realization.

In order to leverage the microscopic realization of marginal deformations in terms of the $\mathcal{N} = 2$ vector multiplet, we use the language of cohomological fields introduced in [10, 11]. A key ingredient is the existence of a Killing vector $v$ which is the square of a supersymmetry transformation. Given $v$ we can define its dual $\kappa = g(v, \bullet)$ and the interior product on forms in the cohomology $\iota_v : \omega \in \Omega^*(\mathcal{M}) \mapsto (\iota_v \omega)(\bullet) := \omega(v, \bullet)$. Left and right handed generalized Killing spinors $\zeta_i$ and $\overline{\chi}^i$ of norm $s(x)$ and $\tilde{s}(x)$ respectively generate the supersymmetry transformations. These functions are related to the Killing vector field as $s\tilde{s} = \|v\|^2$. Using this geometric data one can construct the cohomological fields $\phi = \tilde{s}X + s\overline{X}$ and $\Psi_\mu = \zeta_i \sigma_\mu \overline{\lambda}_{\bar{i}} + \overline{\chi}^i \overline{\sigma}_\mu \lambda_i$ in terms of a standard $\mathcal{N} = 2$ vector multiplet $(X, \lambda_i, A_\mu)$. Then the supersymmetry variations take the form

$$\delta A = i \Psi \tag{3.23}$$

$$\delta \Psi = \iota_v F + i \, d_A \phi \tag{3.24}$$

$$\delta \phi = \iota_v \Psi, \tag{3.25}$$

while the action can be written as

$$S = \frac{1}{\pi g_{\mathrm{YM}}^2} \int_{\mathcal{M}} \Omega \wedge \mathrm{Tr} \left( \phi + \Psi + F \right)^2 + \delta(\dots), \tag{3.26}$$

where $\Omega$ is a $v$-equivariantly closed multiform whose zero-form and two-form components are given by [10]

$$\Omega_0 = \frac{s - \tilde{s}}{s + \tilde{s}} + i \frac{\theta g_{\mathrm{YM}}^2}{8 \pi^2}$$

$$\Omega_2 = -2i \frac{s - \tilde{s}}{(s + \tilde{s})^3} d\kappa - \frac{4i}{(s + \tilde{s})^3} \kappa \wedge d(s - \tilde{s}).$$

$\Omega$ also has a 4-form component which is not needed for the subsequent computations. Since $\Omega$ is equivariantly closed, we see by a straight forward computation that

$$(i\,d + \iota_v)\Omega \wedge \text{Tr}(F + \Psi + \phi)^2 = \delta(\text{Tr}(F + \Psi + \phi)^2), \tag{3.27}$$

showing that up to supersymmetrically exact terms the Lagrangian is equivariantly closed. Following Atiyah-Bott-Berline-Vergne, this implies that the action localizes equivariantly to a sum over fixed-points of the Killing vector [29]. Indeed one can show that modulo $\delta$-exact terms

$$S = 32\tau \sum_{x:s(x)=0} \frac{1}{\varepsilon_x \varepsilon'_x} \mathcal{A}(x) + 32\overline{\tau} \sum_{x:\tilde{s}(x)=0} \frac{1}{\varepsilon_x \varepsilon'_x} \overline{\mathcal{A}}(x), \tag{3.28}$$

with $\varepsilon_x, \varepsilon'_x$ characterizing the manifold close to the fixed point $x$. In deriving (3.28) we used that

$$\text{Tr}X^2 = +8i\,\mathcal{A}(x), \qquad \text{Tr}\overline{X}^2 = -8i\,\overline{\mathcal{A}}(x). \tag{3.29}$$

Let us show with more detail the above argument on the deformed sphere background of [18]. The Killing vector field and the functions $s$ and $\tilde{s}$ are given by

$$v = \frac{1}{\ell}\frac{\partial}{\partial\phi} + \frac{1}{\tilde{\ell}}\frac{\partial}{\partial\chi}, \qquad s = 2\sin^2\left(\frac{\rho}{2}\right), \qquad \tilde{s} = 2\cos^2\left(\frac{\rho}{2}\right). \tag{3.30}$$

The vector field has fixed-points at the north pole ($\rho = 0$) and the south pole ($\rho = \pi$). We define the following multiform

$$\eta = \frac{\kappa}{\|v\|^2} - i\frac{\kappa \wedge d\kappa}{\|v\|^4} \quad \Rightarrow \quad (i\,d + \iota_v)\eta = 1, \tag{3.31}$$

which is well defined everywhere except at the fixed points of $v$. Away from the poles, we can write

$$\begin{aligned}
\Omega \wedge \text{Tr}(\phi + \Psi + F)^2 &= ((i\,d + \iota_v)\eta) \wedge \Omega \wedge \text{Tr}(\phi + \Psi + F)^2 \\
&= (i\,d + \iota_v)\left(\eta \wedge \Omega \wedge \text{Tr}(\phi + \Psi + F)^2\right) + \delta\left(\eta \wedge \Omega \wedge \text{Tr}(\phi + \Psi + F)^2\right).
\end{aligned} \tag{3.32}$$

To use this, we can cut out small balls of radius $\epsilon$ around the poles of the sphere and apply Stokes theorem, giving

$$\begin{aligned}
&\int_{\mathcal{M}} \Omega \wedge \text{Tr}(\phi + \Psi + F)^2 \\
&= \lim_{\epsilon \to 0}\left( i\int_{(S^3_\epsilon(N)\cup S^3_\epsilon(S))} \eta \wedge \Omega \wedge \text{Tr}(\phi + \Psi + F)^2 \right. \\
&\qquad \left. + \delta\left(\int_{\mathcal{M}\backslash(B_\epsilon(N)\cup B_\epsilon(S))} \eta \wedge \Omega \wedge \text{Tr}(\phi + \Psi + F)^2\right)\right). 
\end{aligned} \tag{3.33}$$

Using the definition of $\eta$ in eq. (3.31) and that of $\Omega$ in eq. (3.27) we compute

$$\begin{aligned}
\eta \wedge \Omega \wedge \text{Tr}(\phi + \Psi + F)^2 =\ &\text{Tr}(\phi^2)\frac{-i\,\omega_3}{(s\tilde{s})^2}\kappa \wedge d\kappa + \frac{\omega_1}{s\tilde{s}}\kappa \wedge \text{Tr}(\Psi^2) + 2\frac{\omega_1}{s\tilde{s}}\kappa \wedge \text{Tr}(\phi F) \\
&+ 2\frac{\omega_1}{s\tilde{s}}\kappa \wedge \text{Tr}(\Psi \wedge F)\frac{-2i\,\omega_3}{(s\tilde{s})^2}\kappa \wedge d\kappa \wedge \text{Tr}(\phi\Psi) + \dots,
\end{aligned} \tag{3.34}$$

where we have omitted forms of degree less than three since they do not contribute to the integrals. $\omega_1$ and $\omega_3$ are the coefficients of the one- and three-form in $\eta$

$$\omega_1 = \frac{s-\tilde{s}}{(s+\tilde{s})} + i\frac{\theta g_{YM}^2}{8\pi^2}, \qquad \omega_3 = \frac{(s-\tilde{s})(s^2 + 4s\tilde{s} + \tilde{s}^2)}{(s+\tilde{s})^3} + i\frac{\theta g_{YM}^2}{8\pi^2}. \tag{3.35}$$

Using the explicit form of the Killing one-form, we find that the leading term at small $\epsilon$ for the surface integrals in (3.33) is

$$
\begin{aligned}
i\int_{S_\epsilon^3(N)} \mathrm{Tr}(\phi^2)\omega_3 \kappa \wedge d\kappa &= i\int_{S_\epsilon^3(N)} \mathrm{Tr}(\phi^2)(N)(-i)\frac{1}{\epsilon^4}\omega_3(N)\frac{-2\epsilon}{f}E^1 \wedge E^2 \wedge E^3 \\
&= \frac{-2}{f\epsilon^3}\omega_3(N)\mathrm{Tr}(\phi^2)(N)\int_0^{\pi/2}\int_0^{2\pi}\int_0^{2\pi}\epsilon\ell\cos\theta\,\epsilon\tilde{\ell}\sin\theta\,\epsilon f\,d\phi\,d\chi\,d\theta \\
&= -4\pi^2\ell\tilde{\ell}\,\omega_3(N)\mathrm{Tr}(\phi^2)(N) \\
&= -16\pi^2\ell\tilde{\ell}\left(-1 + \frac{i\theta g_{YM}^2}{8\pi^2}\right)\mathrm{Tr}(X^2)(N) \\
&= 4i\pi g_{YM}^2\tau\ell\tilde{\ell}\,\mathrm{Tr}(X^2)(N).
\end{aligned}
\tag{3.36}
$$

All other terms contributing to the first integral in (3.33) are suppressed by a factor $s\tilde{s} \approx \epsilon^2$ and thus vanish. The computation around the south pole works in the same way. One can also check that terms in the second integral are well-behaved and finite when taking $\epsilon$ to zero. This proves that, modulo $\delta-$exact terms,

$$S = -4i\tau\ell\tilde{\ell}\,\mathrm{Tr}(X^2)(N) + 4i\overline{\tau}\ell\tilde{\ell}\,\mathrm{Tr}(\overline{X}^2)(S) = 32\ell\tilde{\ell}\left(\tau A(N) + \overline{\tau}\overline{A}(S)\right). \tag{3.37}$$

In the presence of multiple marginal deformations (3.37) generalizes to

$$S = 32\ell\tilde{\ell}\sum_i\left(\tau_i A_i(N) + \overline{\tau}_{\bar{i}}\overline{A}_{\bar{i}}(S)\right). \tag{3.38}$$

From this we get

$$\partial_i\overline{\partial}_{\bar{j}}\log Z_{\mathcal{M}} = (32\ell\tilde{\ell})^2\left\langle A_i(N)\overline{A}_{\bar{j}}(S)\right\rangle_{\mathcal{M}}. \tag{3.39}$$

For the case of the round sphere where $\ell = \tilde{\ell} = r$, (3.39) reproduces the result of [15, 16]. Remarkably, (3.39) can be generalized to any *any* supersymmetric background. For a manifold $\mathcal{M}_s$ with many isolated fixed points, (3.39) generalizes to a sum over all fixed points,

$$\partial_i\overline{\partial}_j\log Z_{\mathcal{M}_s} = (32)^2\sum_{x:s(x)=0,y:\tilde{s}(y)=0}\frac{1}{\varepsilon_x\varepsilon_x'\varepsilon_y\varepsilon_y'}\langle\mathcal{A}_i(x)\overline{\mathcal{A}}_{\bar{j}}(y)\rangle, \tag{3.40}$$

where $\ell$ and $\tilde{\ell}$ in (3.39) are replaced by the two equivariant parameters $\varepsilon_x$, $\varepsilon_x'$ ($\varepsilon_y$, $\varepsilon_y'$) that characterize the plus (minus) fixed points of the chosen Killing vector on $\mathcal{M}_s$.

The two-point function appearing on the right hand side of (3.39) is related to the two-point function of marginal operators due to the supersymmetric Ward identities, and hence is proportional to the Zamolodchikov metric.[3] Choosing $\ell = rb, \tilde{\ell} = \frac{r}{b}$, we can parameterize (3.39) as

$$\partial_i\overline{\partial}_{\bar{j}}\log Z_{\mathcal{M}} = \frac{g_{i\bar{j}}}{12}\left(1 + \widetilde{P}(\tau_i, \overline{\tau}_i, b)\right). \tag{3.41}$$

---

[3]One might have expected that since (3.39) contains scalar operators, it could have been written in terms of elementary geometric data of the deformed sphere, *e.g.* the geodesic distance between the two fixed points. However, this is not true because of the presence of various non-trivial background fields which modify the two-point function of microscopic fields in the Lagrangian and consequently the two-point functions of various composite operators. It would be interesting to determine explicitly how the two-point functions depends on such background fields.

We can then integrate this up to

$$\log Z = \frac{K\left(\tau_i, \overline{\tau}_{\overline{i}}\right)}{12}\left(1 + P(\tau_i, \overline{\tau}_i, b)\right) + P_h\left(\tau_i, b\right) + \overline{P}_h\left(\overline{\tau}_{\overline{i}}, b\right), \tag{3.42}$$

where $K(\tau_i, \overline{\tau}_i)$ is the Kähler potential, $\tilde{P} = \frac{1}{\dim_{\mathcal{M}_C}} g^{i\overline{j}} \partial_i \overline{\partial}_{\overline{j}}(KP)$, and $P_h$ and $\overline{P}_h$ are holomorphic and anti-holomorphic functions of the moduli. In the next section we will argue that $P_h(\tau_i, b)$ and $\overline{P}_h(\overline{\tau}_{\overline{i}}, b)$ are Weyl-invariant functionals of the supergravity backgrounds.

## 3.4. The moduli anomaly and the finite part of the free energy

In this section we use insights from the moduli anomaly [14] and extended conformal manifolds [17] to further constrain the form of the free energy for $\mathcal{N} = 2$ SCFTs. In particular we demonstrate that the functions $P_h(\tau_i, b)$ and $\overline{P}_h(\overline{\tau}_{\overline{i}}, b)$ appearing in eq. (3.42) are Weyl-invariant functionals of the supergravity backgrounds. Moreover, we argue that the ambiguous part of the function $K\left(\tau_i, \overline{\tau}_{\overline{i}}\right) P(\tau_i, \overline{\tau}_i, b)$ is proportional to the $\mathrm{I}_{(\mathrm{Weyl})^2}$ term.

We start with the superspace expression of the Weyl-anomaly which involves the Kähler potential [13, 14]:

$$\delta_\Sigma \log Z \supset +\frac{1}{192\pi^2} \int \mathrm{d}^4x \mathrm{d}^4\theta \mathrm{d}^4\overline{\theta}\, \mathcal{E} \left(\delta\Sigma + \delta\overline{\Sigma}\right) K\left(\tau_i, \overline{\tau}_{\overline{i}}\right). \tag{3.43}$$

The normalization is fixed by the two point function of marginal operators which is proportional to the Zamolodchikov metric. After evaluating the right hand side of (3.43) in component form and setting the moduli $\tau_i$ to be constant, it takes the form of a Weyl variation of the supersymmetric Gauss-Bonnet term (see eq. (5.9) in [13] or (2.2) in [14]),

$$\frac{1}{96\pi^2} K\left(\tau_i, \overline{\tau}_{\overline{i}}\right) \int \mathrm{d}^4x\, \delta_\sigma \left(\sqrt{g}\left(\frac{1}{8}E_4 - \frac{1}{12}\Box R + \tilde{c}\left(\tau_i, \overline{\tau}_{\overline{i}}\right)\left(C_{\mu\nu\rho\sigma}C^{\mu\nu\rho\sigma} + \cdots\right)\right)\right), \tag{3.44}$$

where $\tilde{c}\left(\tau_i, \overline{\tau}_{\overline{i}}\right)$ is an arbitrary function of the moduli. The dependence on the Gauss-Bonnet term is unambiguous while the $(\mathrm{Weyl})^2$ term appears with $\tilde{c}\left(\tau_i, \overline{\tau}_{\overline{i}}\right)$ as this is the only Weyl-invariant and supersymmetric local term that can be constructed from $\mathcal{N} = 2$ supergravity background fields. Comparing (3.44) with (3.42) we conclude that $P_h$ and $\overline{P}_h$ are Weyl-invariant, possibly non-local functionals of the supergravity background fields. The free energy, modulo holomorphic Weyl-invariant terms is

$$\frac{K\left(\tau_i, \overline{\tau}_i\right)}{12} + \frac{\mathrm{I}_{(\mathrm{Weyl})^2}}{96\pi^2} K(\tau_i, \overline{\tau}_i)\tilde{c}(\tau_i, \overline{\tau}_i) + \gamma(\tau_i, \overline{\tau}_i, b), \tag{3.45}$$

where $\gamma(\tau_i, \overline{\tau}_i, b)$ is an unambiguous, Weyl-invariant and necessarily non-local functional of the supergravity background.

We finally use the Kähler ambiguity and the choice of possible counter-terms [17] to further constrain $\tilde{c}(\tau_i, \overline{\tau}_i)$. The ambiguities in the free energy must be taken into account by appropriate counter-terms. These ambiguities render the partition function multivalued on the conformal manifold $\mathcal{M}_C$. Including the counter-terms with coupling $t_i$ makes the partition function single-valued on the extended conformal manifold [17] parameterized by marginal couplings and the counter-term couplings. For example on the round sphere the free energy is $\frac{K(\tau_i, \overline{\tau}_i)}{12}$ and the possible supergravity counter-terms are [15,30](see also appendix A of [19])[4]:

$$t_\chi \frac{1}{192\pi^2} \int \mathrm{d}^4x \mathrm{d}^4\theta\, \mathcal{E} \left(\Xi - W^{\alpha\beta}W_{\alpha\beta}\right) + c.c, \qquad t_W \frac{1}{192\pi^2} \int \mathrm{d}^4x \mathrm{d}^4\theta\, \mathcal{E} W^{\alpha\beta}W_{\alpha\beta} + c.c. \tag{3.46}$$

---

[4]$t_\chi$ and $t_W$ have to be holomorphic function of moduli but here we treat them as independent couplings which can have holomorphic ambiguities.

The second term, which is proportional to $I_{(Weyl)^2}$, vanishes for the round sphere and the first term evaluates to

$$-\frac{1}{12}(t_\chi + \overline{t}_\chi)\chi(\mathbb{S}^4) = -\frac{1}{6}(t_\chi + \overline{t}_\chi).\tag{3.47}$$

The free energy is then a well defined function of marginal couplings and $t_\chi$ if the Kähler shift,

$$K \to K + F + \overline{F},\tag{3.48}$$

is accompanied by a shift in the coupling $t_\chi$,

$$t_\chi \to t_\chi + \frac{F}{2}.\tag{3.49}$$

For supergravity backgrounds with a non-zero $I_{(Weyl)^2}$, the Kähler shift must also be accompanied by an appropriate shift in the coupling $t_W$ to make the free energy well-defined. Since the coupling can be shifted by holomorphic functions of moduli only, this constrains the form of $\widetilde{c}(\tau_i, \overline{\tau}_i)$ in eq. (3.45). It must satisfy

$$K(\tau_i, \overline{\tau}_i)\widetilde{c}(\tau_i, \overline{\tau}_i) = \alpha K(\tau_i, \overline{\tau}_i) + \beta(\tau_i, \overline{\tau}_i),\tag{3.50}$$

where $\alpha$ is a constant and $\beta(\tau_i, \overline{\tau_i})$ is an unambiguous function of the moduli which is independent of the Kähler shifts. This also implies that $\beta(\tau_i, \overline{\tau}_i)$ is modular invariant since duality transformations generate Kähler shifts. The free energy is now a well-defined function of the moduli $\tau_i$, the coupling constant for the Gauss-Bonnet term $t_\chi$ and the coupling constant for the $I_{(Weyl)^2}$ term if the Kähler shifts are accompanied by the shifts

$$t_\chi \to t_\chi + \frac{F}{2} \qquad t_W \to t_W - 4\alpha F.\tag{3.51}$$

In summary, the free energy of an $\mathcal{N} = 2$ SCFT on an arbitrary supersymmetry background takes the form

$$\log Z = \frac{K(\tau_i, \overline{\tau}_i)}{12} + \frac{\alpha}{96}K(\tau_i, \overline{\tau}_i)I_{(Weyl)^2} + \frac{1}{96}\beta(\tau_i, \overline{\tau}_i)I_{(Weyl)^2} + \gamma(\tau_i, \overline{\tau}_i, b) + P_h(\tau_i, b) + \overline{P}_h(\overline{\tau}_i, b),\tag{3.52}$$

where $\alpha$ is a theory-dependent constant, $\beta(\tau_i, \overline{\tau}_i)$ and $\gamma(\tau_i, \overline{\tau}_i, b)$ are modular-invariant functions of the moduli and $P_h(\overline{P}_h)$ is a Weyl-invariant, holomorphic (anti-holomorphic) function of the moduli and supergravity background parameters.

## 4. Supersymmetric localization and the free energy on the deformed sphere

In this section we study SCFTs on the specific supergravity background of [18]. We start by reviewing the background fields needed to preserve supersymmetry. We then analyze the partition function of supersymmetric theories on this background. We discuss the structure of the free energy for Abelian $\mathcal{N} = 4$ super Yang-Mills (SYM) as well as $\mathcal{N} = 4$ SYM at large $N$ using localization. We then elucidate the definition of $\mathcal{N} = 4$ SYM on the deformed sphere, showing that the finite part of the free energy is independent of the deformation parameter if one chooses the theory such that near the poles it reduces to $\mathcal{N} = 4$ SYM in the $\Omega$-background, indicating a symmetry enhancement.

### 4.1. Review of the $\mathcal{N} = 2$ supersymmetric background

An $\mathcal{N} = 2$ supersymmetric theory can be coupled to a supergravity background by turning on appropriate background fields in the $\mathcal{N} = 2$ Poincaré supergravity. To preserve supersymmetry one needs to add non-minimal couplings in the Lagrangian. For the vector multiplet the

Lagrangian takes the form [10, 18]

$$\mathscr{L}_{\text{vec}} = \mathscr{L}_{\text{vec}}^{\text{cov}} + \text{Tr}\Big[16B_{mn}(F^+_{mn}\overline{X} + F^-_{mn}X)) - 64B^{+\mu\nu}B_{+\mu\nu}\overline{X}^2 \quad +64B^{-\mu\nu}B_{-\mu\nu}X^2$$
$$-2(D - \frac{R}{3})X\overline{X}\Big], \quad (4.1)$$

where $\mathscr{L}_{\text{vec}}^{\text{cov}}$ is obtained from the flat space Lagrangian by covariantizing all derivatives with respect to the metric and background gauge fields for the $R$-symmetry, and $X$ is the complex scalar field of the vector multiplet. Similarly for the hypermultiplet Lagrangian one needs to add non-minimal couplings with the background two-form, curvature and the scalar field to preserve supersymmetry,

$$\mathscr{L}_{\text{hyp}} = \mathscr{L}_{\text{hyp}}^{\text{cov}} + i B_{\mu\nu} \text{Tr}\Big(\psi^1 \sigma^{\mu\nu}\psi^2 - \overline{\psi}^1 \overline{\sigma}^{\mu\nu}\overline{\psi}^2\Big) - \frac{1}{4}\Big(D - \frac{2}{3}R\Big)\text{Tr}\big(Z_1\overline{Z}_1 + Z_2\overline{Z}_2\big), \quad (4.2)$$

where $\psi_i$ and $Z_i$ are fermions and scalars in the hypermultiplet.

In order to preserve supersymmetry on the deformed sphere with $U(1) \times U(1)$ isometry and the metric in (2.17), one needs to further turn on non-trivial background fields. These were determined in [18], which we reproduce here in a slightly more conventional form[5]. The background two-form field is given by

$$B^+_{\mu\nu} = \frac{1 - \cos^2\frac{\rho}{2}\cos\rho}{16fg}\Big(\big(\cos\theta\,(g-f) + \sin\theta h\big)\big(E^1 \wedge E^4 + E^2 \wedge E^3\big)$$
$$+ \big(\sin\theta\,(g-f) - \cos\theta h\big)\big(E^2 \wedge E^4 + E^3 \wedge E^1\big)\Big)$$
$$B^-_{\mu\nu} = \frac{1 + \sin^2\frac{\rho}{2}\cos\rho}{16fg}\Big(\big(\cos\theta\,(f-g) + h\sin\theta\big)\big(E^1 \wedge E^4 - E^2 \wedge E^3\big)$$
$$+ \big(\sin\theta\,(f-g) - \cos\theta h\big)\big(E^2 \wedge E^4 - E^3 \wedge E^1\big)\Big), \quad (4.3)$$

where $f$, $g$, and $h$ are defined in (2.19). The expression for the background $SU(2)_R$ field is more complicated and requires the explicit expression for the generalized Killing spinor of [18] to solve for it. It can be expressed as

$$\mathcal{V}_\mu \text{d}x^\mu = \mathbf{V}_a E^a, \quad (4.4)$$

where $\mathbf{V}_a$ are $SU(2)$-valued components of the background field in the local frame. These are given by

$$\mathbf{V}_1 = \widetilde{V}_{1,1}\tau^3 + \widetilde{V}_{1,2}\tau^2_{\chi+\phi},$$
$$\mathbf{V}_2 = \widetilde{V}_{2,1}\tau^3 + \widetilde{V}_{2,2}\tau^2_{\chi+\phi},$$
$$\mathbf{V}_3 = \widetilde{V}_{3,3}\tau^1_{\chi+\phi},$$
$$\mathbf{V}_4 = \widetilde{V}_{4,3}\tau^1_{\chi+\phi}, \quad (4.5)$$

where we have defined $\tau^1_{\chi+\phi} = \cos(\chi + \phi)\tau^1 + \sin(\chi + \phi)\tau^2$ and $\tau^2_{\chi+\phi} = \cos(\chi + \phi)\tau^2 - \sin(\chi + \phi)\tau^1$. The terms multiplying the various matrices in (4.5)

---

[5]These background fields are not uniquely determined. Indeed there is a three-parameter family of background fields which preserve supersymmetry on the deformed sphere [10, 11, 18, 23]. In reproducing the background fields in (4.3)-(4.7) we have made a convenient choice of these parameters such that the the background fields are smooth at the poles.

are given by

$$
\begin{aligned}
\widetilde{V}_{1,1} &= \frac{\sin^2\theta}{2f\sin\rho\cos\theta} + \frac{\cos\theta}{2g\sin\rho} + \frac{h\sin\theta\cos\rho}{2fg\sin\rho}\left(1 + \frac{\sin^2\rho}{2}\right) - \frac{1}{2\ell\cos\theta\sin\rho} \\
\widetilde{V}_{1,2} &= \frac{\sin\theta\cos\rho}{2f\sin\rho}\left(1 - \frac{\widetilde{\ell}^2}{gf} + \frac{\sin^2\rho}{2}\left(1 - \frac{f}{g}\right)\right) \\
\widetilde{V}_{2,1} &= \frac{\cos^2\theta}{2f\sin\rho\sin\theta} + \frac{\sin\theta}{2g\sin\rho} - \frac{h\cos\theta\cos\rho}{2fg\sin\rho}\left(1 + \frac{\sin^2\rho}{2}\right) - \frac{1}{2\widetilde{\ell}\sin\theta\sin\rho} \\
\widetilde{V}_{2,2} &= -\frac{\cos\theta\cos\rho}{2f\sin\rho}\left(1 - \frac{\ell^2}{gf} + \frac{\sin^2\rho}{2}\left(1 - \frac{f}{g}\right)\right) \\
\widetilde{V}_{3,3} &= -\frac{\cos\rho}{2f\sin\rho}\left(1 - \frac{\ell^2\widetilde{\ell}^2}{gf^3} + \frac{\sin^2\rho}{2}\left(1 - \frac{f}{g}\right)\right) \\
\widetilde{V}_{4,3} &= \frac{h\cos\rho}{2fg\sin\rho}\left(1 - \frac{\ell^2\widetilde{\ell}^2}{gf^3} + \frac{\sin^2\rho}{2}\left(1 - \frac{f}{g}\right)\right).
\end{aligned}
\tag{4.6}
$$

Finally the expression for the background scalar field, after subtracting the contribution from the curvature coupling, is

$$
\begin{aligned}
D(x) - \frac{R}{3} &= \frac{1}{f^2} - \frac{1}{g^2} + \frac{h^2}{f^2g^2} - \frac{4}{fg} - \frac{\sin^2\rho\cos^2\rho}{4f^2g^2}\left(f^2 + g^2 - 2fg + h^2\right) \\
&+ \left(\frac{1}{g}\partial_\rho - \frac{h}{gf\sin\rho}\partial_\theta + \frac{\ell^2\widetilde{\ell}^2\cos\rho}{gf^4\sin\rho} + \frac{\left(\ell^2 + \widetilde{\ell}^2 - f^2\right)\cos\rho}{gf^2\sin\rho} - \frac{\cos\rho}{f\sin\rho}\right)\left(\frac{1}{f} - \frac{1}{g}\right)\sin\rho\cos\rho \\
&+ \left(\frac{1}{f\sin\rho}\partial_\theta + \frac{\ell^2\widetilde{\ell}^2 h\cos\rho}{g^2f^4\sin\rho} + \frac{2\cot 2\theta}{f\sin\rho} - \frac{h\cos\rho}{fg\sin\rho}\right)\frac{h}{fg}\sin\rho\cos\rho.
\end{aligned}
\tag{4.7}
$$

## 4.2. The localized partition function

In this section we consider corrections to the free energy coming from the squashing of the sphere to an ellipsoid. We will pay close attention to the logarithmic divergence in the free energy as well as its dependence on the marginal couplings of the SCFT. We start by giving the localized partition function for $\mathcal{N}=2$ vector and hypermultiplets and then specialize to the theory with a hypermultiplet in the adjoint representation.

The localized partition function is given by [18]

$$
\mathcal{Z} = \int \mathrm{d}a \left(\prod_{\alpha\in\Delta_+}(a\cdot\alpha)^2\right) e^{-\frac{8\pi^2}{g_{\mathrm{YM}}^2}\frac{\ell\widetilde{\ell}}{r^2}\mathrm{Tr}\,a^2}|Z_{\mathrm{inst}}|^2 Z_{\mathrm{vec}}Z_{\mathrm{hyp}},
\tag{4.8}
$$

where $Z_{\mathrm{vec}}$ and $Z_{\mathrm{hyp}}$ are the one-loop contributions from the vector multiplet and hypermultiplet. $Z_{\mathrm{inst}}$ is the contribution of instantons at the north and the south pole. On the ellipsoid

the one-loop contributions are given by

$$
Z_{\text{vec}} = \left(Z_{\text{vec},\text{U}(1)}\right)^{r_G} \prod_{\alpha \in \Delta_+} \frac{1}{(a \cdot \alpha)^2} \prod_{m,n \geq 0} \left(\left((m+1)b + \frac{n+1}{b}\right)^2 + \frac{\ell\widetilde{\ell}}{r^2}(a \cdot \alpha)^2\right) \quad (4.9)
$$
$$
\left(\left(mb + \frac{n}{b}\right)^2 + \frac{\ell\widetilde{\ell}}{r^2}(a \cdot \alpha)^2\right).
$$

$$
Z_{\text{hyp}} = \prod_{\rho \in \mathcal{R}} \prod_{m,n \geq 0} \left((m+\tfrac{1}{2})b + \frac{n+\tfrac{1}{2}}{b} + i\sqrt{\frac{\ell\widetilde{\ell}}{r^2}}(a \cdot \rho + \mu)\right)^{-1} \quad (4.10)
$$
$$
\left((m+\tfrac{1}{2})b + \frac{n+\tfrac{1}{2}}{b} - i\sqrt{\frac{\ell\widetilde{\ell}}{r^2}}(a \cdot \rho + \mu)\right)^{-1},
$$

where $r_G$ is the rank of the gauge group, the product on $\alpha$ is over all positive roots and the product on $\rho$ is over all weights in the representation $\mathcal{R}$ of the hypermultiplet. The hypermultiplet mass is $\frac{\mu}{r}$. $Z_{\text{vec},\text{U}(1)}$ is the one-loop determinant associated with each element of the Cartan subalgebra, and is given by

$$
Z_{\text{vec},\text{U}(1)} = Q \prod_{m,n \geq 0, (m,n) \neq (0,0)} \left((m+1)b + \frac{n+1}{b}\right)\left(mb + \frac{n}{b}\right), \quad (4.11)
$$

where $Q = b + b^{-1}$. This term is normally dropped on the sphere because it only contributes to an overall constant. However, on more general manifolds it encodes interesting dependence on the background and are needed to reproduce the correct correlators when taking derivatives with respect to the deformation parameter [19]. The instanton contribution is given by the Nekrasov partition function with equivariant parameters $b$, $\frac{1}{b}$ and the hypermultiplet mass $\frac{\mu}{r}$.

Equations (4.10) and (4.11) are divergent and need to be regularized, except for a specific choice of the hypermultiplet where the product of $Z_{\text{vec}}$ and $Z_{\text{hyp}}$ is finite. The regularized one-loop determinants can be expressed in terms of the Upsilon function $\Upsilon_b(x)$ [31,32] which has zeros at $x = mb + \frac{n}{b} + Q, -mb - \frac{n}{b}$ for all non-negative integers $m$ and $n$.

$$
Z_{\text{vec}} = \left(\Upsilon_b'(0)\right)^{r_G} \prod_{\alpha \in \Delta_+} \frac{\Upsilon_b(i\sqrt{\frac{\ell\widetilde{\ell}}{r^2}}a \cdot \alpha)\Upsilon_b(-i\sqrt{\frac{\ell\widetilde{\ell}}{r^2}}a \cdot \alpha)}{(a \cdot \alpha)^2},
$$
$$
Z_{\text{hyp}} = \prod_{\rho \in \mathcal{R}} \left(\Upsilon_b\left(i\sqrt{\frac{\ell\widetilde{\ell}}{r^2}}(a \cdot \rho + \mu) + \frac{Q}{2}\right)\right)^{-1}. \quad (4.12)
$$

### 4.2.1. Abelian $\mathcal{N} = 4$ SYM

We now use the localization results to explore the free energy of $\mathcal{N} = 4$ SYM. Due to the possibility of non-minimal couplings, there is a subtlety in defining what we mean by an $\mathcal{N} = 4$ theory on curved space. In this example we consider the theory of an $\mathcal{N} = 2$ vector multiplet and a massless adjoint hypermultiplet coupled to the curved space by turning on the supergravity background fields in (4.3)-(4.7).

The instanton partition function for the abelian theory is independent of the deformation and is given by [19,33]

$$
Z_{\text{inst},\text{U}(1)} = \prod_{k=1}^{\infty} \frac{1}{1 - e^{2\pi i k \tau}}. \quad (4.13)
$$

The one-loop determinants need to be regularized. On the round sphere the logarithmic divergence in the free energy is given by $-4a \log \Lambda_{\text{UV}} = -\log \Lambda_{\text{UV}}$. After regularizing the infinite products in (4.10) and (4.11) the free energy can be written as

$$\log Z = \left(\frac{Q^2}{4} - 2\right) \log \Lambda_{\text{UV}} - \frac{1}{2} \log(\tau - \overline{\tau}) + \log \Upsilon_b'(0) + \log Z_{\text{inst,U(1)}} + \log \overline{Z_{\text{inst,U(1)}}}. \quad (4.14)$$

Comparing with the general form of the free energy in eq. (3.52) we see that

$$I_{(\text{Weyl})^2} = 64\pi^2 \left(\frac{Q^2}{4} - 1\right), \quad \alpha = \beta(\tau, \overline{\tau}) = P_h(\tau_i, b) = \overline{P}_h(\overline{\tau}_{\overline{i}}, b) = 0,$$
$$\gamma(\tau_i, \overline{\tau}_i, b) = \log \Upsilon_b'(0). \quad (4.15)$$

The first term is theory independent and is computed from the logarithmic divergence of the free energy.

### 4.2.2. $\mathcal{N} = 4$ SYM at large $N$

Let us now consider the large $N$ limit of the $\mathcal{N} = 2$ theory with a massive adjoint hypermultiplet. The instanton contribution can be ignored in this limit and the localized partition function can be written as

$$Z(b, \mu) = \int \prod_i \mathrm{d}\sigma_i \prod_{i<j} (\sigma_{ij})^2 e^{-\frac{8\pi^2}{\lambda} N \sum_i \sigma_i^2} Z_{\text{vec}} Z_{\text{hyp}}, \quad (4.16)$$

where we have set $\ell = rb$ and $\widetilde{\ell} = \frac{r}{b}$. The one-loop determinants can now be written as

$$Z_{\text{vec}} = \Upsilon_b'(0)^{N-1} \prod_{i \neq j} \frac{\Upsilon_b(i\sigma_{ij})}{i\sigma_{ij}},$$
$$Z_{\text{hyp}} = \left(\Upsilon_b(\frac{Q}{2} + i\mu)\right)^{-N+1} \prod_{i \neq j} \left(\Upsilon_b(i\sigma_{ij} + i\mu + \frac{Q}{2})\right)^{-1}. \quad (4.17)$$

After some manipulation the infinite products can be written as

$$\begin{aligned}
Z_{\text{vec}} Z_{\text{hyp}} &= \left(\frac{\Upsilon_b'(0)}{\Upsilon_b(\frac{Q}{2} + i\mu)}\right)^{N-1} \prod_{i \neq j} \prod_{n=1}^{\infty} \prod_{m=1}^{n} \left(1 - \frac{(n-2m)^2 \gamma'^2}{(n+i\sigma_{ij}')^2}\right) \left(1 - \frac{(n-2m+1+i\rho)^2 \gamma'^2}{(n+i\sigma_{ij}')^2}\right)^{-1} \\
&= \left(\frac{\Upsilon_b'(0)}{\Upsilon_b(\frac{Q}{2} + i\mu)}\right)^{N-1} \\
&\quad \times \exp\left(-\sum_{i \neq j} \sum_{p=1}^{\infty} \frac{(\gamma')^{2p}}{p} \sum_{n=1}^{\infty} \left[\frac{1}{(n+i\sigma_{ij}')^{2p}} \sum_{m=1}^{n} \left((n-2m)^{2p} - (n-2m+1+i\rho)^{2p}\right)\right]\right)
\end{aligned} \quad (4.18)$$

where $\gamma' = \sqrt{1 - \frac{4}{Q^2}}$, $\rho = \frac{2\mu}{Q\gamma'}$, and $\sigma_i' = 2\sigma_i/Q$. The sum over $n$ in (4.18) is divergent and needs to be regularized. To do this we cut off the sum at some large $n = r\Lambda_{\text{UV}}'$ and then take the limit $\Lambda_{\text{UV}}' \to \infty$. However, there is a subtlety as to how $\Lambda_{\text{UV}}'$ is chosen since $n$ appears with the redefined fields $\sigma_{ij}'$ in (4.18). In particular, we claim that to match with the definition of $\sigma_{ij}'$, $\Lambda_{\text{UV}}' = 2\Lambda_{\text{UV}}/Q$, where $\Lambda_{\text{UV}}$ is the cutoff that is held fixed as the squashing parameter is varied. While we do not prove it here, we believe that this choice is consistent with the enhancement to $\mathcal{N} = 4$ supersymmetry. We will later show that this is also consistent with the results in [19] for integrated correlators.

Equation (4.18) can then be reexpressed as

$$Z_{\text{vec}} Z_{\text{hyp}} = \left( \frac{\Upsilon_b'(0)}{\Upsilon_b(\frac{Q}{2}+i\mu)} \right)^{N-1} \exp\left( (1+\rho^2) \sum_{i\neq j} \sum_{p=1}^{\infty} (\gamma')^{2p} f_p(\sigma_{ij}',\rho) \right), \tag{4.19}$$

where the functions $f_p(x,\rho)$ can be written in terms of digamma functions and their derivatives, plus a logarithmic divergence. The first few examples are

$$
\begin{aligned}
f_1(x,\rho) &= -\log\Lambda_{\text{UV}}' + \psi(1+ix) + ix\,\psi'(1+ix) \\
f_2(x,\rho)) &= -\log\Lambda_{\text{UV}}' + \psi(1+ix) + 3ix\,\psi'(1+ix) - \left( \frac{1}{4}(1+\rho^2) + \frac{3}{2}x^2 \right) \psi''(1+ix) \\
&\quad - \left( \frac{1}{12}(1+\rho^2)ix + \frac{1}{6}ix^3 \right) \psi'''(1+ix) \\
f_3(x) &= -\log\Lambda_{\text{UV}}' + \psi(1+ix) + 5ix\,\psi'(1+ix) \\
&\quad - \left( \frac{5}{6}(1+\rho^2) + 5x^2 \right) \psi''(1+ix) - \left( \frac{5}{6}(1+\rho^2)ix + \frac{5}{3}ix^3 \right) \psi'''(1+ix) \\
&\quad + \left( \frac{1}{72}(1+\rho^2)(3+\rho^2) + \frac{5}{24}(1+\rho^2)x^2 + \frac{5}{24}x^4 \right) \psi^{(4)}(1+ix) \\
&\quad + \left( \frac{1}{120}(1+\rho^2)(3+\rho^2)ix + \frac{1}{72}(1+\rho^2)ix^3 + \frac{1}{120}ix^5 \right) \psi^{(5)}(1+ix).
\end{aligned}
\tag{4.20}
$$

Combining the divergent part of $f_p$ with the divergent part of the prefactor in (4.19) we find that the free energy has the logarithmic divergence

$$\log Z \supset -\left( \frac{Q^2}{4} - 1 + \mu^2 \right) \left( N^2 - 1 \right) \log\Lambda_{\text{UV}}'. \tag{4.21}$$

Using the functions in (4.20) one can then find corrections to the free energy order by order in $\gamma'$. We are particularly interested in the situation where $\lambda \gg 1$. In this case we expect generic eigenvalues in the matrix model to be widely separated from each other at the saddle point. In other words one has that $|\sigma_{ij}'| \gg 1$ for generic $i$ and $j$. In this case we have that all $f_p(x,\rho)$ satisfy $f_p(x,\rho) \approx -\log\Lambda_{\text{UV}}' + \log(ix)$. Hence, after regularization, which removes the $\Lambda_{\text{UV}}$ dependence, we have that[6]

$$Z_{\text{vec}} Z_{\text{hyp}}\Big|_{\text{reg.}} \approx \prod_{i\neq j} \left( \frac{Q}{2} \right)^{\frac{Q^2}{2}-2+2\mu^2} \exp\left( (1+\rho^2) \frac{(\gamma')^2}{1-(\gamma')^2} \left[ \log\left( i\sigma_{ij}' \right) \right] \right) = \prod_{i\neq j} (i\sigma_{ij})^{\frac{Q^2}{4}-1+\mu^2}. \tag{4.22}$$

Substituting this into (4.16) we find that the partition function is

$$Z\Big|_{\text{reg.}} \approx \int \prod_i d\sigma_i \prod_{i<j} (\sigma_{ij}^2)^{\frac{Q^2}{4}+\mu^2} e^{-\frac{8\pi^2}{\lambda}N\sum_i \sigma_i^2}. \tag{4.23}$$

This is very close to the form of a Gaussian matrix model. In fact if $\mu = \pm i\frac{Q\gamma'}{2}$ it *is* the Gaussian matrix model, as we know it must be since at these points $Z_{\text{vec}} Z_{\text{hyp}} = 1$. In the large $N$ limit the saddle point equation is

$$\frac{16\pi^2}{\lambda} N\sigma_i = 2\left( \frac{Q^2}{4} + \mu^2 \right) \sum_{j\neq i} \frac{1}{\sigma_i - \sigma_j}, \tag{4.24}$$

---

[6]We have dropped the prefactor in (4.19) as it does not contribute to the leading order result at large $N$.

which is equivalent to the saddle point equation for a Gaussian matrix model with $\lambda$ replaced by $\left(\frac{Q^2}{4} + \mu^2\right)\lambda$. Hence, the free energy is

$$\log Z\big|_{\text{reg.}} \approx \frac{N^2}{2}\left(\frac{Q^2}{4} + \mu^2\right)\log\left(\lambda\left(\frac{Q^2}{4} + \mu^2\right)\right). \tag{4.25}$$

Note that (4.25) is very similar to the free energy for strongly coupled $\mathcal{N} = 2^*$ on the round sphere [34–36].

Due to the logarithmic divergence, one needs combinations of at least three derivatives with respect to $Q$ and $\mu$ for scheme independence. For $\mu = 0$ we can write the free energy, including the divergent part, as

$$\log Z \approx N^2\left[-\left(\frac{Q^2}{4} - 2\right)\log\Lambda_{\text{UV}} + \frac{Q^2}{8}\log\lambda + \frac{Q^2}{8}\log\frac{Q^2}{4}\right]. \tag{4.26}$$

Comparing this with the Weyl anomaly and the general form of the finite part in (3.52), we see that in the large $N$ limit[7]

$$\text{I}_{(\text{Weyl})^2} = \frac{1}{64\pi^2}\left(\frac{Q^2}{4} - 1\right), \qquad \alpha = 512, \qquad \beta(\tau_i, \overline{\tau}_i) = P_h(\tau_i, b) = \overline{P}_h(\overline{\tau}_{\overline{i}}, b) = 0,$$

$$\gamma(\tau_i, \overline{\tau}_i, b) = -\frac{N^2 Q^2}{8}\log\frac{Q^2}{4}. \tag{4.27}$$

### 4.3. Subtleties regarding $\mathcal{N} = 4$ on curved space and an infinite set of relations

It is a subtle issue to identify a quantum field theory on a general curved space as a CFT. For a conformally flat background, one can canonically map a CFT from flat space to the curved space. For a generic curved space one cannot unambiguously determine a unique fixed point due to the presence of more than one length scale. The ambiguity manifests itself in various choices for the non-minimal coupling of the QFT to the curved space. Demanding supersymmetry substantially restricts these choices but still leaves some ambiguity. One possibility is to determine the beta function of the theory on the curved space[8]. However, the renormalization group flow is not well understood on curved space [39].

This ambiguity is also present when we place $\mathcal{N} = 4$ SYM on a curved space. We define the $\mathcal{N} = 4$ theory as $\mathcal{N} = 2^*$ with a special value of the hypermultiplet mass parameter, $\mu/r$. Naïvely, one would associate an adjoint hypermultiplet with $\mu = 0$ to the $\mathcal{N} = 4$ theory. Indeed, this turns out to be the correct choice for the theory on the round sphere [40]. Near the poles, the round supersymmetric background is equivalent to the $\Omega$-background with equivariant parameters $\epsilon_1 = \epsilon_2 = \frac{1}{r}$. For generic values of the hypermultiplet mass superconformal symmetry is broken and only 8 supercharges are preserved. For the correct value of the adjoint hypermultiplet mass, all 32 superconformal symmetries are restored in the $\Omega$-background. This value depends on the equivariant parameters, and for the round sphere corresponds to $\mu = 0$.

The $\mathcal{N} = 4$ value of $\mu$ is modified on the deformed sphere. At the superconformal point, the mass parameter $m_N$ which appears in the Nekrasov partition function is equal to either equivariant parameter [40]. In the more general case, $m_N$ is related to the hypermultiplet mass $\frac{\mu}{r}$ and the equivariant parameters $\epsilon_1, \epsilon_2$ as [40]

$$m_N = i\frac{\mu}{r} + \frac{\epsilon_1 + \epsilon_2}{2}. \tag{4.28}$$

---

[7]We use that in the large $N$ limit $K(\tau, \overline{\tau}) = -6N^2\log\lambda$.

[8]The flat space beta function can be determined from the localized partition function by examining the scale dependence of the one-loop determinants and comparing it to the classical part [33, 37, 38] for the case of the sphere.

Since the equivariant parameters are equal on the round sphere, setting $\mu = 0$ leads to $m_N = \frac{1}{r}$, which is the conformal mass term on the round sphere. For the deformed sphere we have that $\epsilon_1 = \frac{b}{r}$, $\epsilon_2 = \frac{1}{br}$, and thus get[9]

$$i\mu = \pm\frac{1}{2}\left(b - \frac{1}{b}\right) \tag{4.29}$$

so that $m_N = \epsilon_1$ or $m_N = \epsilon_2$ at this pole. The relation in (4.29) is also the value advocated in [41] by demanding that the instanton partition function is trivial. By embedding the supersymmetric background in $\mathcal{N} = 4$ supergravity [42], one can show that the supersymmetry enhances at the poles [43] for this particular value of mass parameter. Moreover, for this value of the hypermultiplet mass the infinite products in the one-loop determinants simplify to $Z_{\text{vec}}Z_{\text{hyp}} = 1$[10]. Consequently, the partition function with any gauge group is independent of the deformation and given by

$$Z\left(b, \mu = \pm\left(\frac{i\,b}{2} - \frac{i}{2b}\right)\right) = \int da \exp\left(-\frac{8\pi^2}{g_{\text{YM}}^2}\text{Tr}\, a^2\right)\prod_{\alpha \in \Delta_+}(a \cdot \alpha)^2. \tag{4.30}$$

For the $SU(N)$ gauge group in the large $N$ limit, the result in eq. (4.25) becomes exact for the $\mathcal{N} = 4$ value for the mass and we obtain the free energy

$$\log Z = -\frac{N^2}{2}\log\lambda. \tag{4.31}$$

This has remarkable consequences for the integrated correlators that can be computed by taking derivatives of the free energy with respect to the squashing parameter. In particular

$$\partial_b^n \log Z\left(b, \mu = \pm\left(\frac{i\,b}{2} - \frac{i}{2b}\right)\right) = 0 \tag{4.32}$$

gives a relation between various integrated $n$- and lower-point correlation functions. In [19], three non-trivial relations between various four-point correlators in $\mathcal{N} = 4$ SYM were derived by studying the mass-deformed $\mathcal{N} = 2^*$ on the deformed sphere. We can demonstrate how two of these relations are a simple consequence of (4.32)[11]. To do so, we write the deformed Lagrangian schematically as

$$\mathcal{L} = \frac{1}{g_{\text{YM}}^2}\sum_{n=0}^{\infty}(b-1)^n\left(\mathcal{L}^{0,n} + \mu\mathcal{L}^{1,n} + \mu^2\mathcal{L}^{2,n}\right). \tag{4.33}$$

The first relation in [19] is

$$-64\pi^2\partial_\tau\partial_{\overline{\tau}}\left(\partial_\mu^2 - \partial_b^2\right)\log Z(b, \mu)\Big|_{\mu=0, b=1} = 0. \tag{4.34}$$

---

[9]In computing the partition function via localization only values of the background fields near the poles appear in the one-loop determinants. It is possible that there is a non-trivial profile of the hypermultiplet mass which enhances the symmetry and has the value in (4.29) at the poles.

[10]Another interesting choice is $i\mu = -\frac{1}{2}\left(b + \frac{1}{b}\right)$ for which the one-loop determinants cancel the Vandermonde determinant and only the instantons contribute to the partition function. For $b = 1$ this choice coincides with the theory pointed out in eq. (5.16) of [33].

[11]To be precise, one obtains unambiguous relations between various derivatives of the free energy which can be related to integrated correlators. In doing so one needs to be careful in dealing with possible contributions from redundant operators. We thank G. Festuccia for pointing this out.

This is equivalent to [12]

$$
\begin{aligned}
&-2 \int \prod_{i=1}^{2} \mathrm{d}^4 x_i \sqrt{g(x_i)} \langle \mathscr{L}^{1,0}(x_1)\mathscr{L}^{1,0}(x_2) + 2\mathscr{L}^{0,0}(x_1)\mathscr{L}^{2,0}(x_2) \\
&\qquad - \mathscr{L}^{0,1}(x_1)\mathscr{L}^{0,1}(x_2) - 2\mathscr{L}^{0,0}(x_1)\mathscr{L}^{0,2}(x_2)\rangle \\
&+ \frac{4}{g_{\mathrm{YM}}^2} \int \prod_{i=1}^{3} \mathrm{d}^4 x_i \sqrt{g(x_i)} \langle \mathscr{L}^{0,0}(x_1)\mathscr{L}^{1,0}(x_2)\mathscr{L}^{1,0}(x_3) - \mathscr{L}^{0,0}(x_1)\mathscr{L}^{0,1}(x_2)\mathscr{L}^{0,1}(x_3)\rangle \\
&- \frac{1}{g_{\mathrm{YM}}^4} \int \prod_{i=1}^{4} \mathrm{d}^4 x_i \sqrt{g(x_i)} \langle \mathscr{L}^{0,0}(x_1)\mathscr{L}^{0,0}(x_2)\mathscr{L}^{1,0}(x_3)\mathscr{L}^{1,0}(x_4) \\
&\qquad - \mathscr{L}^{0,0}(x_1)\mathscr{L}^{0,0}(x_2)\mathscr{L}^{0,1}(x_3)\mathscr{L}^{0,1}(x_4)\rangle \\
&= 0\,.
\end{aligned}
\tag{4.35}
$$

It is straightforward to check that the same constraint follows from the deformation independence of the $\mathcal{N} = 4$ theory. In particular, the combination appearing in (4.35) is equal to

$$
-32\pi^2 \partial_\tau \partial_{\bar\tau} \partial_b^2 \left[ \log Z(b, \frac{i\,b}{2} - \frac{i}{2b}) + \log Z(b, -\frac{i\,b}{2} + \frac{i}{2b}) \right]\Bigg|_{b=1}.
\tag{4.36}
$$

Similarly we can show that the second relation in [19],

$$
\left( -6\partial_b^2 \partial_\mu^2 + \partial_\mu^4 + \partial_b^4 - 15\partial_b^2 \right) \log Z(b,\mu) \Big|_{\mu=0,b=1} = 0\,,
\tag{4.37}
$$

is equivalent to

$$
\left( \partial_b^4 - 15\partial_b^2 \right) \left[ \log Z(b, \frac{i\,b}{2} - \frac{i}{2b}) + \log Z(b, -\frac{i\,b}{2} + \frac{i}{2b}) \right]\Bigg|_{b=1} = 0,
\tag{4.38}
$$

where we also used $\log Z(b,\mu) = \log Z(b^{-1},\mu)$, which is evident from the construction of the partititon function.

One can, in fact, derive an infinite number of relations between various integrated correlators using

$$
\sum_n a_n \partial_b^n \left[ \log Z(b, \frac{i\,b}{2} - \frac{i}{2b}) + \log Z(b, -\frac{i\,b}{2} + \frac{i}{2b}) \right]\Bigg|_{b=1} = 0,
\tag{4.39}
$$

where the $a_n$ are chosen to satisfy

$$
\sum_n a_n \partial_b^n (b + \frac{1}{b})^2 \Big|_{b=1} = 0,
\tag{4.40}
$$

in order to ensure that ambiguous terms in the free energy do not contribute. For example, the above relation with the operator $\partial_b^5 + 6\partial_b^4$ translates into

$$
\left( \partial_b^5 + 6\partial_b^4 - 10\partial_b^3 \partial_\mu^2 - 6\partial_b^2 \partial_\mu^2 - 4\partial_\mu^4 \right) \log Z(b,\mu) \Big|_{b=1,\mu=0} = 0.
\tag{4.41}
$$

The third relation in [19] is

$$
-16c = (3\partial_b^2 \partial_\mu^2 - \partial_\mu^4 - 16\tau_2^2 \partial_\tau \partial_{\bar\tau} \partial_\mu^2) \log Z(b,\mu) \Big|_{b=1,\mu=0},
\tag{4.42}
$$

where $c = \frac{N^2-1}{4}$. The authors of [19] provided overwhelming evidence for (4.42), and it is straightforward to show that this is consistent with the large-$N$ expression given in (4.25), but it does not follow from (4.32) alone. It would be interesting to demonstrate (4.42) directly.

---

[12]We set $\theta = 0$ in the following expressions.

## Acknowledgements

We thank A. Ardehali for helpful discussions and G. Festuccia for helpful discussions and comments on the manuscript. This research is supported in part by Vetenskapsrådet under grant #2016-03503 and by the Knut and Alice Wallenberg Foundation under grant Dnr KAW 2015.0083. JAM thanks the Center for Theoretical Physics at MIT for kind (virtual) hospitality during the course of this work.

## A. Ward Identities for two-point functions

In this section we use the notations and the flatspace $\mathcal{N} = 2$ supersymmetry transformations of the stress-tensor multiplet from [25]. For convenience, the transformations we use are

$$\delta O_2 = i\overline{\chi}_{\dot{\alpha}A}\overline{\xi}^{\dot{\alpha}A} + i\xi_A^\alpha \chi_\alpha^A \tag{A.1}$$

$$\delta \chi_\alpha^A = H_\alpha{}^\beta \xi_\beta^I + \dots \tag{A.2}$$

$$\delta \overline{\chi}_{\dot{\alpha}A} = -\partial_{\alpha\dot{\alpha}} O_2 \xi_A^\alpha + \dots \tag{A.3}$$

$$\delta \overline{H}{}^{\dot{\beta}}{}_{\dot{\alpha}} = -\frac{i}{2}\overline{J}_{\alpha\dot{\alpha}}{}^{\dot{\beta}A}\xi_A^\alpha - \frac{2i}{3}\left(\partial_{\alpha\dot{\alpha}}\overline{\chi}^{\dot{\beta}A} + \partial_\alpha{}^{\dot{\beta}}\overline{\chi}_{\dot{\alpha}}^A\right)\xi_A^\alpha \tag{A.4}$$

$$\delta j_{\alpha\dot{\alpha}} = -\frac{i}{2}J_{\alpha\dot{\alpha}\beta}{}^A\xi_A^\beta + \dots \tag{A.5}$$

$$\delta t_{\alpha\dot{\alpha}A}{}^B = iJ_{\alpha\dot{\alpha}\beta}{}^B\xi_A^\beta + \dots - \frac{1}{2}\delta_A^B\left[iJ_{\alpha\dot{\alpha}\beta}{}^C\xi_C^\beta + \dots\right] \tag{A.6}$$

$$\delta J_{\alpha\dot{\alpha}\beta}{}^A = \frac{2}{3}\left(\partial_{\alpha\dot{\alpha}}H_\beta{}^\gamma + \partial_{\beta\dot{\alpha}}H_\alpha{}^\gamma\right)\xi_\gamma^A - 2\partial_{\gamma\dot{\alpha}}H_\beta{}^\gamma\xi_\alpha^A - 2\partial_{\gamma\dot{\alpha}}H_\alpha{}^\gamma\xi_\beta^A + \dots \tag{A.7}$$

$$\begin{aligned}\delta \overline{J}_{\alpha\dot{\alpha}\dot{\beta}A} =& -2T_{\alpha\dot{\alpha}\beta\dot{\beta}}\xi_A^\beta - \xi_A^\beta\left(\frac{2}{3}\partial_{\alpha\dot{\alpha}}j_{\beta\dot{\beta}} - \frac{1}{3}\partial_{\alpha\dot{\beta}}j_{\beta\dot{\alpha}} - \partial_{\beta\dot{\alpha}}j_{\alpha\dot{\beta}}\right)\\ &+ 2\xi_B^\beta\left(\frac{2}{3}\partial_{\alpha\dot{\alpha}}t_{\beta\dot{\beta}A}{}^B - \frac{1}{3}\partial_{\alpha\dot{\beta}}t_{\beta\dot{\alpha}A}{}^B - \partial_{\beta\dot{\alpha}}t_{\alpha\dot{\beta}A}{}^B\right) + \dots\end{aligned} \tag{A.8}$$

$$\delta T_{\alpha\dot{\alpha}\beta\dot{\beta}} = \frac{i}{4}\xi_A^\gamma\left(2\partial_{\gamma\dot{\alpha}}J_{\beta\dot{\beta}\alpha}{}^A + 2\partial_{\gamma\dot{\beta}}J_{\alpha\dot{\alpha}\beta}{}^A - \partial_{\alpha\dot{\alpha}}J_{\beta\dot{\beta}\gamma}{}^A - \partial_{\beta\dot{\beta}}J_{\alpha\dot{\alpha}\gamma}{}^A\right) + \dots \tag{A.9}$$

The ellipses denote terms that are not necessary for the following computations as the corresponding terms are easily seen to drop out. The parameter $\xi$ is assumed to be Grassmann odd.

We make use of the following Ward identities

$$0 = \delta \left\langle T_{\alpha\dot\alpha\beta\dot\beta} \overline{J}_{\gamma\dot\gamma\dot\delta A} \right\rangle \tag{A.10}$$

$$= -2 \left\langle T_{\alpha\dot\alpha\beta\dot\beta} T_{\gamma\dot\gamma\delta\dot\delta} \right\rangle \xi_A^\delta + \frac{i}{4} \xi_B^\delta \left\langle \left( 2\partial_{\gamma\dot\alpha} J_{\beta\dot\beta\alpha}{}^A + 2\partial_{\gamma\dot\beta} J_{\alpha\dot\alpha\beta}{}^A - \partial_{\alpha\dot\alpha} J_{\beta\dot\beta\gamma}{}^A - \partial_{\beta\dot\beta} J_{\alpha\dot\alpha\gamma}{}^A \right) \overline{J}_{\gamma\dot\gamma\dot\delta A} \right\rangle \tag{A.11}$$

$$0 = \delta \left\langle J_{\alpha\dot\alpha\beta}{}^A \overline{H}^{\dot\epsilon}{}_{\dot\delta} \right\rangle \tag{A.12}$$

$$= -\frac{i}{2} \left\langle J_{\alpha\dot\alpha\beta}{}^A \overline{J}_{\gamma\dot\delta}{}^{\dot\epsilon B} \right\rangle \xi_B^\gamma + \left\langle \left[ \frac{2}{3} \left( \partial_{\alpha\dot\alpha} H_\beta{}^\gamma + \partial_{\beta\dot\alpha} H_\alpha{}^\gamma \right) \xi_\gamma^A - 2\partial_{\gamma\dot\alpha} H_\beta{}^\gamma \xi_\alpha^A - 2\partial_{\gamma\dot\alpha} H_\alpha{}^\gamma \xi_\beta^A \right] \overline{H}^{\dot\epsilon}{}_{\dot\delta} \right\rangle \tag{A.13}$$

$$0 = \delta \left\langle t_{\alpha\dot\alpha A}{}^B \overline{J}_{\beta\dot\beta\dot\gamma C} \right\rangle \tag{A.14}$$

$$= -i\xi_A^\delta \left\langle J_{\alpha\dot\alpha\delta}{}^B \overline{J}_{\beta\dot\beta\dot\gamma C} \right\rangle + \frac{i}{2} \delta_A^B \xi_K^\delta \left\langle J_{\alpha\dot\alpha\delta}{}^K \overline{J}_{\beta\dot\beta\dot\gamma C} \right\rangle$$
$$+ 2\xi_D^\delta \left\langle t_{\alpha\dot\alpha A}{}^B \left( \frac{2}{3} \partial_{\beta\dot\beta} t_{\delta\dot\gamma C}{}^D - \frac{1}{3} \partial_{\beta\dot\gamma} t_{\delta\dot\beta C}{}^D - \partial_{\delta\dot\beta} t_{\beta\dot\gamma C}{}^D \right) \right\rangle \tag{A.15}$$

$$0 = \delta \left\langle j_{\alpha\dot\alpha} \overline{J}_{\beta\dot\beta\dot\gamma C} \right\rangle \tag{A.16}$$

$$= \frac{i}{2} \xi_B^\delta \left\langle J_{\alpha\dot\alpha\delta}{}^B \overline{J}_{\beta\dot\beta\dot\gamma C} \right\rangle - \xi_C^\delta \left\langle j_{\alpha\dot\alpha} \left( \frac{2}{3} \partial_{\beta\dot\beta} j_{\delta\dot\gamma} - \frac{1}{3} \partial_{\beta\dot\gamma} j_{\delta\dot\beta} - \partial_{\delta\dot\beta} j_{\beta\dot\gamma} \right) \right\rangle \tag{A.17}$$

$$0 = \delta \left\langle O_2 \overline\chi_{\dot\alpha A} \right\rangle = i\xi_B^\alpha \left\langle \chi_\alpha^B \overline\chi_{\dot\alpha A} \right\rangle - \left\langle O_2 \partial_{\alpha\dot\alpha} O_2 \right\rangle \xi_A^\alpha \tag{A.18}$$

$$0 = \delta \left\langle \chi_\alpha^A \overline{H}^{\dot\gamma}{}_{\dot\delta} \right\rangle = \left\langle H_\alpha{}^\beta \overline{H}^{\dot\gamma}{}_{\dot\delta} \right\rangle \xi_\beta^A - \frac{2i}{3} \left\langle \chi_\alpha^A \left( \partial_{\beta\dot\delta} \overline\chi^{\dot\gamma B} + \partial_\beta{}^{\dot\gamma} \overline\chi_{\dot\delta}{}^B \right) \right\rangle \xi_B^\beta. \tag{A.19}$$

Combing the first two of these we find

$$-2 \left\langle T_{\alpha\dot\alpha\beta\dot\beta} T_{\gamma\dot\gamma}{}^{\dot\delta\delta} \right\rangle \xi_\delta^A \xi_A^{'\gamma} \tag{A.20}$$

$$= \frac{1}{2} \xi_B^\delta \left[ 2\partial_{\delta\dot\alpha} \left\langle \left( \frac{2}{3} \left[ \partial_{\beta\dot\beta} H_\alpha{}^\epsilon + \partial_{\alpha\dot\beta} H_\beta{}^\epsilon \right] \xi_\epsilon^{'A} - 2\partial_{\epsilon\dot\beta} H_\alpha{}^\epsilon \xi_\beta^A - 2\partial_{\epsilon\dot\beta} H_\beta{}^\gamma \xi_\alpha^{'A} \right) \overline{H}^{\dot\delta}{}_{\dot\gamma} \right\rangle \right.$$
$$+ 2\partial_{\delta\dot\beta} \left\langle \left( \frac{2}{3} \left[ \partial_{\alpha\dot\alpha} H_\beta{}^\epsilon + \partial_{\beta\dot\alpha} H_\alpha{}^\epsilon \right] \xi_\epsilon^{'A} - 2\partial_{\epsilon\dot\alpha} H_\beta{}^\epsilon \xi_\alpha^A - 2\partial_{\epsilon\dot\alpha} H_\alpha{}^\gamma \xi_\beta^{'A} \right) \overline{H}^{\dot\delta}{}_{\dot\gamma} \right\rangle$$
$$- \partial_{\alpha\dot\alpha} \left\langle \left( \frac{2}{3} \left[ \partial_{\beta\dot\beta} H_\delta{}^\epsilon + \partial_{\delta\dot\beta} H_\beta{}^\epsilon \right] \xi_\epsilon^{'A} - 2\partial_{\epsilon\dot\beta} H_\delta{}^\epsilon \xi_\beta^A - 2\partial_{\epsilon\dot\beta} H_\beta{}^\gamma \xi_\delta^{'A} \right) \overline{H}^{\dot\delta}{}_{\dot\gamma} \right\rangle$$
$$- \partial_{\beta\dot\beta} \left. \left\langle \left( \frac{2}{3} \left[ \partial_{\alpha\dot\alpha} H_\delta{}^\epsilon + \partial_{\delta\dot\alpha} H_\alpha{}^\epsilon \right] \xi_\epsilon^{'A} - 2\partial_{\epsilon\dot\alpha} H_\delta{}^\epsilon \xi_\alpha^A - 2\partial_{\epsilon\dot\alpha} H_\alpha{}^\gamma \xi_\delta^{'A} \right) \overline{H}^{\dot\delta}{}_{\dot\gamma} \right\rangle \right]. \tag{A.21}$$

Combing the first with the third and fourth

$$-2\xi_B^{'\alpha} \left\langle T_{\alpha\dot\alpha\beta\dot\beta} T_{\gamma\dot\gamma}{}^{\dot\delta\delta} \right\rangle \xi_\delta^A = -\xi_C^\alpha \xi_\delta^{'D} \left\langle \partial_{\alpha\dot\alpha} t_{\beta\dot\beta B}{}^C \left( \frac{2}{3} \partial_{\gamma\dot\gamma} t^{\dot\delta\delta}{}_D{}^A - \frac{1}{3} \partial_\gamma{}^{\dot\delta} t_{\dot\gamma}{}^\delta{}_D{}^A - \partial^\delta{}_{\dot\gamma} t^{\dot\delta}{}_{\gamma D}{}^A \right) \right\rangle$$
$$- \xi_C^\alpha \xi_\delta^{'D} \left\langle \partial_{\alpha\dot\beta} t_{\beta\dot\alpha B}{}^C \left( \frac{2}{3} \partial_{\gamma\dot\gamma} t^{\dot\delta\delta}{}_D{}^A - \frac{1}{3} \partial_\gamma{}^{\dot\delta} t_{\dot\gamma}{}^\delta{}_D{}^A - \partial^\delta{}_{\dot\gamma} t^{\dot\delta}{}_{\gamma D}{}^A \right) \right\rangle$$
$$+ \frac{1}{2} \xi_B^\alpha \xi_\delta^{'A} \left\langle \partial_{\alpha\dot\alpha} j_{\beta\dot\beta} \left( \frac{2}{3} \partial_{\gamma\dot\gamma} j^{\dot\delta\delta} - \frac{1}{3} \partial_\gamma{}^{\dot\delta} j_{\dot\gamma}{}^\delta - \partial^\delta{}_{\dot\gamma} j^{\dot\delta}{}_\gamma \right) \right\rangle$$
$$+ \frac{1}{2} \xi_B^{'\alpha} \xi_\delta^A \left\langle \partial_{\alpha\dot\alpha} j_{\beta\dot\beta} \left( \frac{2}{3} \partial_{\gamma\dot\gamma} j^{\dot\delta\delta} - \frac{1}{3} \partial_\gamma{}^{\dot\delta} j_{\dot\gamma}{}^\delta - \partial^\delta{}_{\dot\gamma} j^{\dot\delta}{}_\gamma \right) \right\rangle$$
$$+ \frac{1}{2} \xi_B^\alpha \xi_\delta^{'A} \left\langle \partial_{\alpha\dot\beta} j_{\beta\dot\alpha} \left( \frac{2}{3} \partial_{\gamma\dot\gamma} j^{\dot\delta\delta} - \frac{1}{3} \partial_\gamma{}^{\dot\delta} j_{\dot\gamma}{}^\delta - \partial^\delta{}_{\dot\gamma} j^{\dot\delta}{}_\gamma \right) \right\rangle$$
$$+ \frac{1}{2} \xi_B^{'\alpha} \xi_\delta^A \left\langle \partial_{\alpha\dot\beta} j_{\beta\dot\alpha} \left( \frac{2}{3} \partial_{\gamma\dot\gamma} j^{\dot\delta\delta} - \frac{1}{3} \partial_\gamma{}^{\dot\delta} j_{\dot\gamma}{}^\delta - \partial^\delta{}_{\dot\gamma} j^{\dot\delta}{}_\gamma \right) \right\rangle. \tag{A.22}$$

And the last two yield

$$\xi_A'^\alpha \left\langle H_\alpha{}^\beta \overline{H}{}^{\dot\gamma}{}_{\dot\delta} \right\rangle \xi_\beta^A = \frac{2}{3} \left\langle O_2 \left( \partial_{\beta\dot\delta} \partial^{\dot\gamma\alpha} O_2 + \partial_\beta{}^{\dot\gamma} \partial_{\dot\delta}{}^\alpha O_2 \right) \right\rangle \xi_\alpha'^B \xi_B^\beta. \tag{A.23}$$

From the quantum numbers (including scaling dimension, spin and R-spin) we know that the different 2-point functions take the form

$$\left\langle T_{\alpha\dot\alpha\beta\dot\beta} T_{\gamma\dot\gamma}{}^{\dot\delta\delta} \right\rangle = \frac{1}{V_{\mathbb{S}^{d-1}}^2} (\sigma^\mu)_{\alpha\dot\alpha} (\sigma^\nu)_{\beta\dot\beta} (\sigma^\rho)_{\gamma\dot\gamma} (\sigma^\tau)^{\dot\delta\delta} \frac{C_T}{|x|^8} I_{\mu\nu,\rho\tau}(x) \tag{A.24}$$

$$\left\langle H_\alpha{}^\beta \overline{H}{}^{\dot\alpha}{}_{\dot\beta} \right\rangle = \frac{1}{V_{\mathbb{S}^{d-1}}^2} \frac{C_H x^\mu x^\nu}{2|x|^8} \left( (\sigma_\mu)_{\alpha\dot\beta} (\sigma_\nu)^{\dot\alpha\beta} + (\sigma_\mu)_\alpha{}^{\dot\alpha} (\sigma_\nu)^\beta{}_{\dot\beta} \right) \tag{A.25}$$

$$\left\langle j_{\alpha\dot\alpha} j_{\beta\dot\beta} \right\rangle = \frac{1}{V_{\mathbb{S}^{d-1}}^2} (\sigma^\mu)_{\alpha\dot\alpha} (\sigma^\nu)_{\beta\dot\beta} \frac{C_j}{|x|^6} I_{\mu\nu}(x) \tag{A.26}$$

$$\left\langle t_{\alpha\dot\alpha A}{}^B t_{\beta\dot\beta C}{}^D \right\rangle = \frac{1}{V_{\mathbb{S}^{d-1}}^2} (\sigma^\mu)_{\alpha\dot\alpha} (\sigma^\nu)_{\beta\dot\beta} \frac{C_t}{|x|^6} I_{\mu\nu}(x) \left( \delta_A^D \delta_C^B + \epsilon_{AC} \epsilon^{BD} \right) \tag{A.27}$$

$$\left\langle O_2 O_2 \right\rangle = \frac{1}{V_{\mathbb{S}^{d-1}}^2} \frac{C_O}{|x|^4}. \tag{A.28}$$

Plugging these into the relations we got from the Ward identities we get that

$$C_H = \frac{3}{40} C_T \tag{A.29}$$

$$C_j = -\frac{3}{40} C_T \tag{A.30}$$

$$C_t = -\frac{3}{80} C_T \tag{A.31}$$

$$C_O = \frac{1}{32} C_H = \frac{3}{1280} C_T \tag{A.32}$$

has to hold for the equations to be consistent. We double-checked the last relation by also computing the ratios $C_O/C_t$, $C_O/C_j$ using corresponding Ward identities.

The two-form correlator can be rewritten with spatial indices. To do this we contract (A.25) with appropriate sigma matrices to trade spinor indices for space indices, which, after some simplification gives eq. (3.13).

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
