# Peer review of "Conformal field theories on deformed spheres, anomalies, and supersymmetry"

_SciPost Physics, doi:SciPost Phys. 10, 063 (2021)_

## Round 1 · Referee Report · Anonymous (Referee 1) · 2021-1-21

Strengths
1- Important novel formula for the dependence of partition function of SCFT's in deformed spheres in terms of marginal couplings
Weaknesses
1- Some points in section 4.3 need clarification.
Report
deformed spheres, with the main focus on N=2 supersymmetric CFT's.
The authors derive a new very interesting formula describing the dependence of the free energy
on the marginal couplings of the theory, in particular, the relation to the K\" ahler potential.
The new general formula (3.52) is then checked by specializing to N=2 theories on the ellipsoid, where one can compute the free energy explicitly by using supersymmetric localization.
In the last part, the paper considers the case of N=2 theory with a massive adjoint
hypermultiplet, pointing out that, for a special value of the hypermultiplet mass,
the theory simplifies: the instanton partition function becomes trivial and the one-loop
determinant cancels out. The resulting theory is called $N=4_2$ theory. The
"triviality" of the free energy of $N=4_2$ theory implies an infinite number of identities
involving correlation functions of integrated operators in N=4 theory (some of these
are consistent with results that already appeared in the literature).
In the case $b=1$, this theory seems to coincide with the theory pointed out by Pestun
in eq. (5.13) in [33]. The resulting partition function still has a non-trivial dependence on the couppling from the sum over instanton sectors. Maybe the authors should clarify
the discrepancy with their eq. (4.30).
The $N=4_2$ theory is here used as a tool to derive constraints in the N=4 theory, but otherwise it remains obscure.
It would be useful for readers if the authors also add more explanations on the expected structure
of the $N=4_2$ theory, in particular, which sectors of correlators are expected to differ from those of N=4 theory.
The authors should consider the above points before the paper is accepted for publication.

---

## Round 1 · Referee Report · Anonymous (Referee 2) · 2021-2-5

Strengths
2-new general result for supersymmetric 4d N=2 partition functions on deformed sphere (and other similar backgrounds)
Weaknesses
Report
Section 4 deals with one main example, N=2 SU(N) with one massive adjoint, which is the N=4 SYM in the UV. It would be useful if the authors could comment on why, apparently, m=0 is not the proper SCFT point "on curved space". The confusion is that at m=0 we have an SCFT, and the fact that I couple it to an arbitrary background does not change the fact that it's a CFT. Presumably, it has to do with the precise form of the Pestun-like background, which modify the UV behavior at the poles, introducing some sort of UV contact terms? (If so, the issue is already present in the flat-space Omega background.) Some clarification would be appreciated.
Finally, using that claimed conformal value of the mass, they derive some identities amongst correlators at large N which follows from their general formula. This reproduces and generalizes recent results in the literature.
In conclusion, this clearly written paper presents new, interesting and topical results. I recommend it for publication in SciPost after a small revision addressing the question above.
Requested changes
-Small typo to be corrected in intro: on p3, ref. to (3.39) is presumably to (3.40).
-comment on why m=0 is not the conformal value.

---

## Round 1 · Referee Report · Anonymous (Referee 3) · 2021-2-7

Strengths
2 - Interesting new formula for the partition function of four-dimensional $\mathcal{N}=2$ SCFT's on supersymmetric backgrounds, with a focus on deformed four-spheres.
Weaknesses
Report
The results are general, rigorous and interesting. The paper is clearly written, especially Sections 1-3. I recommend publication in SciPost after the minor requested changes below have been addressed.
Requested changes
-
It may be useful to specify that the variations $\delta\alpha$, $\delta\sigma$ are assumed independent of the coordinates in Eq. (3.2).
-
In Section 4, it is not entirely clear which parts are a review of known results and which parts are original. The authors may add some comments clarifying this.
-
Minor typos: page 3 Zamalodchikov $\to$ Zamolodchikov; position of index $\nu$ in last term of Eq. (3.9); repeated "any" in the text under (3.39).

---

## Round 2 · Referee Report · Anonymous (Referee 3) · 2021-2-16

Report

The authors have addressed my remarks in a satisfactory way. I recommend publication in SciPost.

---

## Round 2 · Referee Report · Anonymous (Referee 1) · 2021-2-24

Report

The authors have incorporated the necessary changes and clarifications.
I now recommend this paper for publication in SciPost.

---

## Round 2 · Author Response

We thank the three referees for their careful reading of the manuscript and for valuable suggestions. We have made several changes in the new version following the points made by the referees. These changes, are listed in the the appropriate section of our resubmission page.

---

## Round 2 · List of Changes

1.On page 3 we changed the reference from eq. (3.39) to eq. (3.40). 2. We added a footnote before eq. (3.2) clarifying that $\delta\sigma$ and $\delta\alpha$ are space independent. 3. We have added extra references in section 4 to clarify what parts of the material were already known. 4. We fixed the spellings of Zamolodchikov. 5. We deleted the repeated “any” in the text after eq. (3.39). 6. We have modified language to avoid the confusion about $m=0$ being the conformal value of the mass. For $m=0$ only eight supercharges are preserved at the poles while for m= I/2 (b-1/b), all 32 supercharges are preserved. We now call this the $\mathcal{N}=4$ value. 7. We added a footnote on page.27 addressing one of the referees’ comments regarding the theory pointed out by Pestun in eq. (5.13) of his paper. 8. We have dropped the names ${\mathcal{N}=4_1$ and $\mathcal{N}=4_2$ as we believe that this causes unnecessary confusion.

Apart from the above changes directly related to referee’s suggestions we have taken the opportunity to fix various typos and to add further clarifications which are summarized below:

  1. We added a few sentences before eq. (4.19) to clarify the subtlety regarding the cutoff. This leads to minor changes in equations (4.20)-(4.23) and (4.25)-(4.27).
  2. We added a third relation between various fourth-derivatives of the free energy in eq. (4.42), which agrees with our large-N results.

---

## Editorial Decision

published